# Regulation of Nodal signaling propagation by receptor interactions and positive feedback

Hannes Preiß[1†], Anna C Kögler[1,2*†], David Mörsdorf[1†‡], Daniel Čapek[1,2], Gary H Soh[1], Katherine W Rogers[1§], Hernán Morales-Navarrete[2], María Almuedo-Castillo[1#], Patrick Müller[1,2*]

[1]Friedrich Miescher Laboratory of the Max Planck Society, Tübingen, Germany; [2]University of Konstanz, Konstanz, Germany

**\*For correspondence:**
anna.koegler@uni-konstanz.de (ACK);
patrick.mueller@uni-konstanz.de (PM)

[†]These authors contributed equally to this work

**Present address:** [‡]Department of Neurosciences and Developmental Biology, University of Vienna, Vienna, Austria; [§]Eunice Kennedy Shriver National Institute of Child Health and Human Development, Bethesda, United States; [#]Centro Andaluz de Biología de Desarrollo, Seville, Spain

**Competing interest:** The authors declare that no competing interests exist.

**Abstract** During vertebrate embryogenesis, the germ layers are patterned by secreted Nodal signals. In the classical model, Nodals elicit signaling by binding to a complex comprising Type I/II Activin receptors (Acvr) and the co-receptor Tdgf1. However, it is currently unclear whether receptor binding can also affect the distribution of Nodals themselves through the embryo, and it is unknown which of the putative Acvr paralogs mediate Nodal signaling in zebrafish. Here, we characterize three Type I (Acvr1) and four Type II (Acvr2) homologs and show that – except for Acvr1c – all receptor-encoding transcripts are maternally deposited and present during zebrafish embryogenesis. We generated mutants and used them together with combinatorial morpholino knockdown and CRISPR F0 knockout (KO) approaches to assess compound loss-of-function phenotypes. We discovered that the Acvr2 homologs function partly redundantly and partially independently of Nodal to pattern the early zebrafish embryo, whereas the Type I receptors Acvr1b-a and Acvr1b-b redundantly act as major mediators of Nodal signaling. By combining quantitative analyses with expression manipulations, we found that feedback-regulated Type I receptors and co-receptors can directly influence the diffusion and distribution of Nodals, providing a mechanism for the spatial restriction of Nodal signaling during germ layer patterning.

## Editor's evaluation

While the Nodal signaling pathway plays essential roles in germ layer induction and patterning during development, the precise function of different zebrafish Nodal receptors had not been determined. Here, the authors identify seven genes encoding Type I and Type II Nodal receptors in zebrafish and use a combination of genetic approaches to show that the Type I receptors Acvr1b-a and Acvr1b-b act redundantly as major mediators of Nodal signaling. Whereas the Acvr2 Type II receptors function partly redundantly and partially independently of Nodal in embryo patterning. Moreover, Type I receptor and co-receptor levels can modulate Nodal distribution, providing a mechanism for the spatial restriction of Nodal signaling during embryo patterning.

## Introduction

The formation of the body plan during early embryogenesis depends on the interplay between evolutionarily conserved signaling pathways. The TGF-β superfamily member Nodal is one of the key regulators of vertebrate development and is required to specify mesoderm and endoderm (collectively termed mesendoderm) during germ layer formation (*Schier, 2009*). In the classical model, Nodal ligands signal through a receptor complex comprising Type I and Type II single-transmembrane

**eLife digest** Building a body is complicated. Cells must organise themselves head-to-tail, belly-to-back, and inside-to-outside. They do this by laying down a chemical map, which is made up of gradients of molecular signals, high in some places and lower in others. The amount of signal each cell receives helps to decide which part of the body it will become.

One of the essential signals in developing vertebrates is Nodal. It helps cells to tell inside from outside and left from right. Cells detect Nodal using an activin receptor and co-receptor complex, which catch hold of passing Nodal proteins and transmit developmental signals into cells. An important model to study Nodal signals is the zebrafish embryo, but the identity of the activin receptors and their exact role in this organism has been unclear.

To find out more, Preiß, Kögler, Mörsdorf et al. studied the activin receptors Acvr1 and Acvr2 in zebrafish embryos. The experiments revealed that two putative Acvr1 and four Acvr2 receptors were present during early development. To better understand their roles, Preiß et al. eliminated them one at a time, and in combination. Losing single activin receptors had no effect. But losing both Acvr1 receptors together stopped Nodal signalling and changed the distribution of the Nodal gradient. Loss of all Acvr2 receptors also caused developmental problems, but they were partly independent of Nodal. This suggests that Acvr1s seem to be able to transmit signals and to shape the Nodal gradient, and that Acvr2s might have another, so far unknown, role.

Nodal signals guide the development of all vertebrates. Understanding how they work in a model species like zebrafish could shed light on their role in other species, including humans. A clearer picture could help to uncover what happens at a molecular level when development goes wrong.

serine/threonine kinase receptors (*Attisano and Wrana, 2002*; *Shi and Massagué, 2003*; *Figure 1A*). Unlike other members of the TGF-β superfamily, Nodal signaling additionally requires the presence of an EGF-CFC co-receptor to activate signaling. Our current understanding of Nodal signaling is that Nodal directly binds to Type II receptors and the EGF-CFC co-receptor Tdgf1, which in turn mediates the recruitment of the Type I receptors. Upon oligomerization of the receptor complex, Type II receptors phosphorylate the Type I receptors in their GS domains, leading to the recruitment and phosphorylation of the C-terminal SSXS motif of the receptor-regulated Smad (R-Smad) proteins Smad2 and Smad3 by the Type I receptor. The activated pSmad2/pSmad3 proteins associate with the co-factor Smad4 and translocate into the nucleus, where they activate target gene expression (*Hill, 2018*; *Macías-Silva et al., 1996*; *Shi and Massagué, 2003*; *Yeo and Whitman, 2001*; *Figure 1A*).

In zebrafish, mesendoderm patterning depends on the two secreted Nodal signals Squint (Sqt) and Cyclops (Cyc) (*Dougan et al., 2003*; *Rogers and Müller, 2019*; *Schier, 2009*; *Shen, 2007*). Nodal expression begins in the yolk syncytial layer at the embryonic margin during the blastula stage and then spreads into the embryo, generating a Nodal signaling gradient. This gradient is translated into different mesendodermal cell fates depending on the signaling level and target gene induction kinetics (*Dubrulle et al., 2015*). Loss of Nodal signaling causes absence of endoderm as well as trunk and head mesoderm, which leads to cyclopia due to a failure to separate the eye fields, resulting in embryonic lethality (*Dubrulle et al., 2015*; *Feldman et al., 1998*; *Gritsman et al., 1999*). Nodal signaling is antagonized by the secreted long-range feedback inhibitor Lefty, which is also produced at the margin (*Meno et al., 1999*; *Thisse and Thisse, 1999*). Establishment and maintenance of a correct signaling range is crucial for correct development, as also excess Nodal signaling – for example in *lefty* mutants – can cause severe patterning defects and embryonic lethality (*Almuedo-Castillo et al., 2018*; *Rogers et al., 2017*).

Measurements of active GFP-tagged fusions showed that Squint and Cyclops proteins have a lower effective diffusivity than their inhibitors Lefty1 (Lft1) and Lefty2 (Lft2) (*Müller et al., 2012*; *Rogers and Müller, 2019*). It has been proposed that this mobility difference is due to interactions between Nodal ligands and membrane-bound diffusion regulators, whereas Lefty proteins move more freely in the extracellular space (*Müller et al., 2012*; *Müller et al., 2013*). Indeed, Nodal's signaling range and distribution dramatically increase in the absence of the zebrafish Tdgf1 co-receptor homolog Oep (*Lord et al., 2021*), and single-molecule imaging has recently shown that the fraction of molecules in the bound state is larger for Nodal than for Lefty (*Kuhn et al., 2022*). Since Nodals strongly bind to

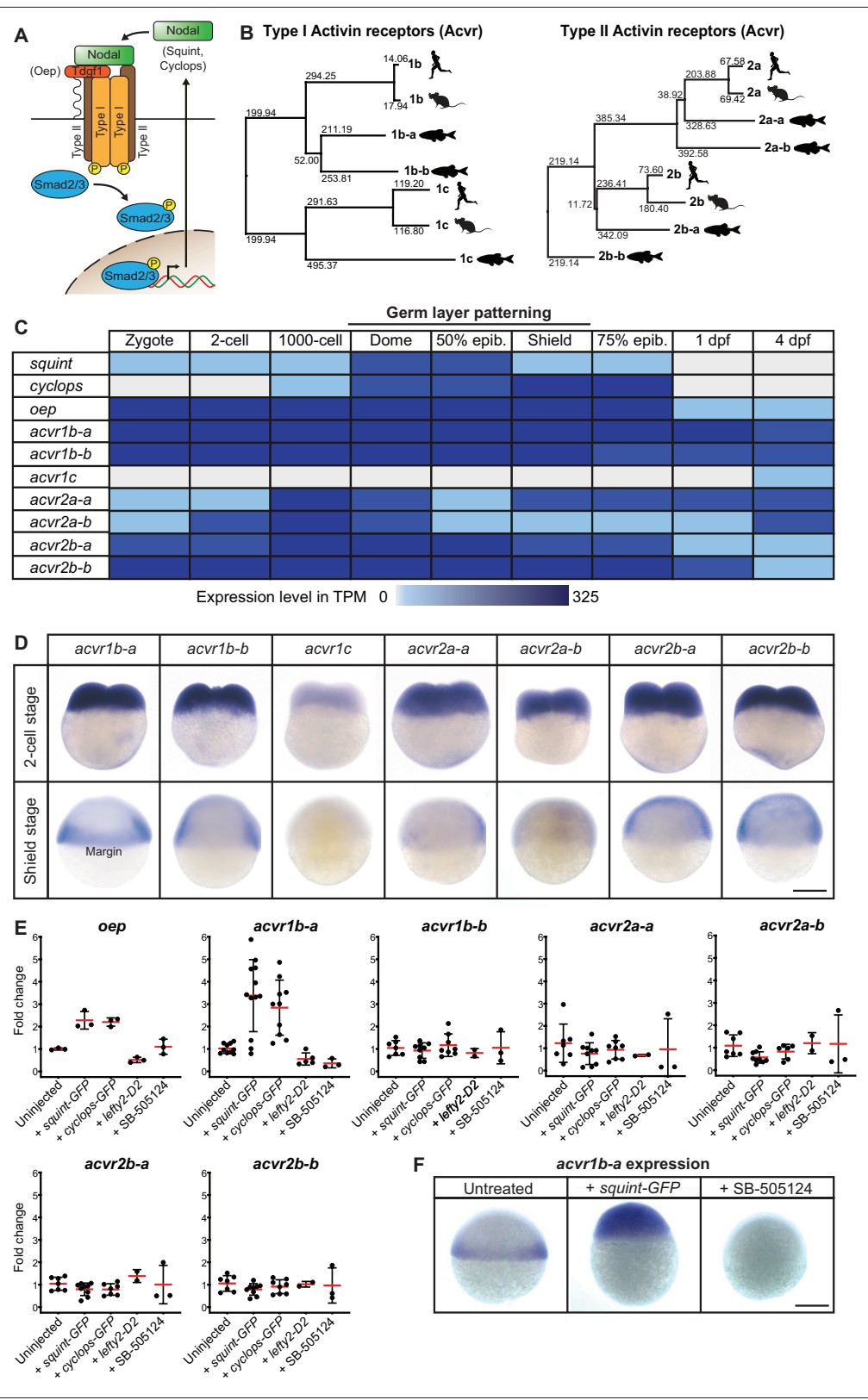

**Figure 1.** Multiple Nodal receptor candidates are expressed during early zebrafish development. (**A**) In the classical model, Nodal signaling requires the recruitment of a receptor complex comprising the co-receptor Oep (Tdgf1 homolog) as well as Type I and Type II Activin receptors (Acvr) to induce phosphorylation and nuclear translocation of the signal transducer pSmad2/3 for the induction of Nodal target genes. (**B**) Phylogenetic

*Figure 1 continued on next page*

*Figure 1 continued*

neighbor-joining alignment tree of Type I and Type II receptor protein sequences from human, mouse, and zebrafish. Bootstrap values are listed at the nodes and indicate evolutionary distances. (**C**) Temporal expression analysis of putative Nodal receptors at different developmental stages. TPM: Transcripts per million. dpf: day(s) post-fertilization. Data adapted from *White et al., 2017*. (**D**) Spatial expression analysis of Type I and Type II receptors at 2 cell and shield stages revealed by in situ hybridization. Except for *acvr1c*, all receptor-encoding transcripts are maternally deposited. At shield stage, *acvr1b-a* is the only receptor that is not uniformly expressed but restricted to the embryonic margin. (**E**) Nodal signaling controls the expression of *acvr1b-a* and *oep*. Fold change of Nodal receptor expression calculated from qRT-PCR experiments comparing the overexpression of 30 pg *squint-GFP* mRNA, 30 pg *cyclops-GFP* mRNA, 30 pg *lefty2-Dendra2* mRNA and exposure to 50 μM SB-505124 Nodal inhibitor to untreated embryos at 6 hours post-fertilization (hpf). Each point is the mean fold change of an individual embryo compared to an untreated embryo. Error bars represent standard deviation. (**F**) In situ hybridization analysis of *acvr1b-a* with increased (*+squint* GFP) or decreased (+SB-505124) Nodal signaling. Scale bar represents 250 μm. See the *Figure 1—source data 1* file for source data and sample size.

The online version of this article includes the following source data and figure supplement(s) for figure 1:

**Source data 1.** Source data for *Figure 1*.

**Figure supplement 1.** Sequences and protein domains of putative Nodal receptors.

the zebrafish Type II receptor Acvr2b-a in vivo (*Wang et al., 2016*), the main Nodal receptors themselves might also act as diffusion regulators. However, it is unclear whether this strong ligand-receptor interaction indeed influences Nodal dispersal, whether receptor binding affects Nodal diffusion or stability in the embryo, and what role other putative Type I and Type II Acvr receptors play in the propagation of Nodal signaling through the embryo.

The two mouse, frog and human Type I receptors Acvr1b (also known as Alk4/TARAM-A) and Acvr1c (also known as Alk7) and the two Type II receptors Acvr2a and Acvr2b were identified using in vitro binding and target induction assays, and cause developmental defects when mutated (*Gritsman et al., 1999*; *Gu et al., 1998*; *Kosaki et al., 1999*; *Matzuk et al., 1995*; *Oh and Li, 1997*; *Reissmann et al., 2001*). Surprisingly, except for the zebrafish co-receptor Oep (*Gritsman et al., 1999*), no zebrafish Nodal receptor mutants are known to recapitulate Nodal loss-of-function phenotypes; and although zebrafish are widely used to investigate Nodal signaling during development, it is unknown which of the receptor paralogs mediate endogenous Nodal signaling during germ layer formation.

To understand the role of the zebrafish receptor homologs in Nodal distribution and signaling, we generated several loss-of-function mutants and used them together with combinatorial morpholino knockdown and CRISPR F0 knockout (KO) approaches to assess compound loss-of-function phenotypes. Due to the severity of single receptor knock-outs in mice (*Gu et al., 1998*; *Kosaki et al., 1999*; *Matzuk et al., 1995*; *Oh and Li, 1997*; *Reissmann et al., 2001*), we expected phenotypes similar to Nodal loss-of-function mutants in zebrafish. Strikingly, loss of individual receptor function did not cause obvious patterning defects. Severe patterning phenotypes and embryonic lethality were observed with combinatorial loss of putative Type II Acvr receptors, but the defects were at least partly independently of Nodal signaling. Instead, only the combined loss of the Type I receptors *acvr1b-a* and *acvr1b-b* phenocopied known Nodal loss-of-function phenotypes (*Dubrulle et al., 2015*; *Feldman et al., 1998*; *Gritsman et al., 1999*), identifying these receptors as the main Type I receptors that mediate early Nodal signaling in zebrafish. Using quantitative imaging assays, we found that Type I receptor and co-receptor levels can modulate Nodal mobility and thereby directly influence the distribution of Nodal in the embryo, providing a mechanism for the spatial restriction of Nodal signaling during germ layer patterning.

## Results
## Nodal Type I and Type II receptors have several putative paralogs in zebrafish

To systematically identify and characterize zebrafish Nodal receptors, we used the protein sequences of the human and mouse Type I receptors Acvr1b and Acvr1c as well as the Type II receptors Acvr2a and Acvr2b as queries for homology searches in the Uniprot database. In addition to the previously experimentally identified zebrafish Type I (*Renucci et al., 1996*) and Type II (*Garg et al., 1999*; *Nagaso*

*et al., 1999*) Nodal receptor orthologs Acvr1b-a, Acvr1c, Acvr2a-a, and Acvr2b-a, our analysis yielded further potential Nodal receptor paralogous sequences named Acvr1b-b, Acvr2a-b and Acvr2b-b (*Funkenstein et al., 2012*; *Li et al., 2019*), respectively, resulting in a total of three putative Type I and four putative Type II receptors. For Acvr2a-b, the start of the gene's coding sequence and the full amino acid sequence of the receptor was not resolved, and corresponding predictions differed between genome assemblies. Therefore, we performed 5'RACE (Rapid Amplification of cDNA Ends) and thereby mapped the start of *acvr2a-b* to chrUn_KN150226v1 (see *Materials and methods* and *Figure 1—figure supplement 1A* for further details). Reconstruction of a putative phylogenetic tree shows a close clustering of the zebrafish receptors with their human and mouse paralogs, and the highest sequence similarity was found between the zebrafish Type I receptors Acvr1b-a and Acvr1b-b (*Figure 1B*). All putative homologs have the typical features of Type I and Type II receptors, including a signal peptide, TGF-β receptor domain, transmembrane domain, cytosolic kinase domain and a GS domain in case of the Type I receptors (*Figure 1—figure supplement 1B-H*).

## Most Nodal receptor paralog transcripts are present during mesendoderm formation

To determine which of the putative Nodal receptor paralogs might have roles in germ layer patterning, we first assessed their expression during early embryogenesis, focusing on early blastula and gastrula stages during which germ layer patterning takes place (*Figure 1C*). Analysis of a published developmental transcriptome (*White et al., 2017*) indicated that the transcripts of most receptor paralogs are present at these stages and before the maternal-zygotic transition, suggesting that they are maternally deposited (*Figure 1C*). Expression of the identified receptors persists throughout larval development up to 4 days post-fertilization (dpf). The only receptor-encoding gene that does not seem to be expressed during early development is the Type I receptor homolog *acvr1c*, which is first detected at 4 dpf (*Figure 1C*). Therefore, all putative receptors except for *acvr1c* are expressed at the developmental stages, during which Nodal signaling patterns the germ layers.

We next used in situ hybridization analysis to characterize the spatial expression patterns of the putative receptors, and in particular to determine whether they are expressed at the embryonic margin, where Nodal signaling induces mesendoderm (*Figure 1D*). In agreement with the temporal analysis (*Figure 1C*), we found that transcripts of all putative Nodal Type I and II receptors – with the exception of *acvr1c* – are evenly distributed at the two-cell stage (*Figure 1D*), consistent with maternal deposition. During early gastrulation (shield stage), most receptors are ubiquitously expressed throughout the embryo (*Garg et al., 1999*; *Nagaso et al., 1999*) – except for *acvr1c*, which is not expressed, and *acvr1b-a*, which is constrained to the embryonic margin (*Figure 1D*), similar to the co-receptor *oep* (*Renucci et al., 1996*; *Vopalensky et al., 2018*). Together, our analyses show that, except for *acvr1c* all putative receptors are expressed at the right time and place to potentially act as mediators of endogenous Nodal signaling during zebrafish germ layer patterning.

## Nodal signaling upregulates *acvr1b-a* expression but does not affect other putative Nodal receptors

In zebrafish, Nodal signaling induces several of its own signaling pathway components, including *squint, cyclops, lefty1, lefty2,* and *oep* (*Bennett et al., 2007*; *Dubrulle et al., 2015*; *Feldman et al., 2002*; *Meno et al., 1999*). To systematically assess potential receptor induction by Nodal signaling, we used qRT-PCR to measure receptor expression levels in embryos with increased Nodal signaling (injection of *squint-GFP* or *cyclops-GFP* mRNA [*Müller et al., 2012*]) or decreased Nodal signaling (injection of *lefty2-Dendra2* mRNA [*Müller et al., 2012*] or treatment with the Nodal inhibitor SB-505124 [*DaCosta Byfield et al., 2004*]). *acvr1c* was excluded from this analysis because its spatiotemporal expression suggests that it does not mediate endogenous Nodal signaling during germ layer patterning (*Figure 1D*). In agreement with previous studies (*Dubrulle et al., 2015*), *oep* and *acvr1b-a* were upregulated by increased Nodal signaling and downregulated by decreased signaling (*Figure 1E*). Upon Nodal overexpression, *acvr1b-a* expression expanded beyond its usual domain at the margin, whereas Nodal inhibition abolished its expression (*Figure 1F*). In contrast, none of the other putative Nodal receptor-encoding genes exhibited a substantial change in expression upon Nodal overexpression or inhibition (*Figure 1E*).

## The Acvr2 receptors act in part redundantly to pattern the early zebrafish embryo through a partially Nodal-independent mechanism

To elucidate the roles of the putative Nodal receptors in germ layer formation, we assessed the effect of their loss of function on embryonic morphologies, starting with the putative Type II receptors Acvr2a-a, Acvr2a-b, Acvr2b-a, and Acvr2b-b. These receptors are thought, and in the case of Acvr2b-a have been shown (*Wang et al., 2016*), to be bound by Nodals directly. To achieve receptor loss of function, we used three different approaches: (1) mutants, (2) morpholino-mediated knockdown (*El-Brolosy et al., 2019*; *Rossi et al., 2015*), and (3) CRISPR-mediated F0 knockouts (KO) (*Kroll et al., 2021*).

To study receptor loss of function in mutants, we obtained *acvr2a-a*$^{SA34654}$ and *acvr2a-b*$^{SA18285}$ mutants from the European Zebrafish Resource Center (EZRC) and generated a mutant allele for *acvr2b-a* using CRISPR/Cas9. In all three mutants, the signal peptide or receptor domain and all downstream domains were disrupted, likely causing a complete loss of gene function (*Figure 2—figure supplement 1A-D*). However, surprisingly, none of the maternal-zygotic homozygous receptor mutants displayed obvious patterning defects at 1 dpf (*Figure 2—figure supplement 1A-D*) and all were viable. This is in stark contrast to the mouse Type IIB receptor mutants that exhibit severe malformations during early embryonic development (*Gu et al., 1998*; *Oh and Li, 1997*).

Genetic compensation can mask potential patterning defects in mutants (*El-Brolosy et al., 2019*; *Rossi et al., 2015*) and may occur in some instances in the *acvr2* mutants (*Figure 2—figure supplement 1E, F*). Therefore, we also assessed the effect of acutely knocking down gene activity using antisense morpholino oligonucleotides targeting the ATG start codons or splice sites of the putative receptor mRNAs. We found that morpholinos targeting individual receptor-encoding mRNAs had no effects or caused non-specific head or tail defects similar to a standard control morpholino (*Figure 2—figure supplement 1G-L*). Some morpholino treatments at high doses increased lethality (*Figure 2—figure supplement 1H-L*), but none of the conditions led to the Nodal-specific patterning defects observed in loss-of-function mutants of other Nodal signaling pathway components (*Dubrulle et al., 2015*; *Feldman et al., 1998*; *Gritsman et al., 1999*).

To complement our mutant and morpholino analyses, we acutely knocked down the putative Type II Nodal receptors using CRISPR-mediated F0 KO (*Kroll et al., 2021*). In this approach, multiple exons of a single gene are targeted by Cas9/gRNA ribonucleoproteins (RNPs), thereby generating a biallelic zygotic KO directly in the injected embryos, the so-called F0 generation. Using this technique, we found that F0 KO of *acvr2a-a*, *acvr2a-b*, *acvr2b-a*, or *acvr2b-b* did not affect embryo morphology (*Figure 2B*), confirming that the disruption of individual Acvr2 receptors has no substantial influence on embryonic patterning.

Teleosts like zebrafish have undergone an additional genome duplication following the two vertebrate-specific rounds of whole-genome duplications (*Meyer and Van de Peer, 2005*), and partial redundancy of paralogs can underlie the lack of abnormal phenotypes in single mutants (*Feldman et al., 1998*; *Leerberg et al., 2019*). To test whether the putative Nodal Type II receptors function redundantly, we used the CRISPR F0 KO strategy to simultaneously disrupt multiple *acvr2* genes, thereby creating combinatorial KOs. Double, triple, and quadruple F0 KO embryos were analyzed for patterning defects. In contrast to the Nodal-specific loss-of-function phenotype, embryos of all KO combinations developed two properly spaced eyes (*Figure 2A*). However, with an increasing number of *acvr2* genes disrupted, embryos displayed more severe dorsalization phenotypes (*Figure 2A and B*): *acvr2b-a*,*acvr2b-b* double KO embryos had mild dorsalization phenotypes with the ventral tail fin partly or completely missing, similar to class 1 (C1) BMP mutant phenotypes (*Kishimoto et al., 1997*). Triple *acvr2a-a*,*acvr2b-a*,*acvr2b-b* F0 KO embryos additionally showed a bent or kinked tail and a thickened yolk extension (*Figure 2A and B*). Embryos disrupted zygotically for *acvr2a-b*,*acvr2b-a*,*acvr2b-b* or all four *acvr2* genes showed the most severe patterning defects, including a kinked or curled tail, the absence of fin tissue, a larger yolk sac with shortened yolk extension, prominent hatching glands, smaller eyes and head structures, cuboidal instead of chevron-shaped somites, head necrosis (*Figure 2A and B*) and lethality at 2–3 dpf. These robust phenotypes were seen in the large majority (97%) of quadruple F0 KO embryos, and although they include typical features of dorsalization, like a curled tail seen upon loss of BMP signaling (*Kishimoto et al., 1997*; *Little and Mullins, 2009*), other aspects like the reduction in head and eye size do not fit the classical dorsalization phenotype.

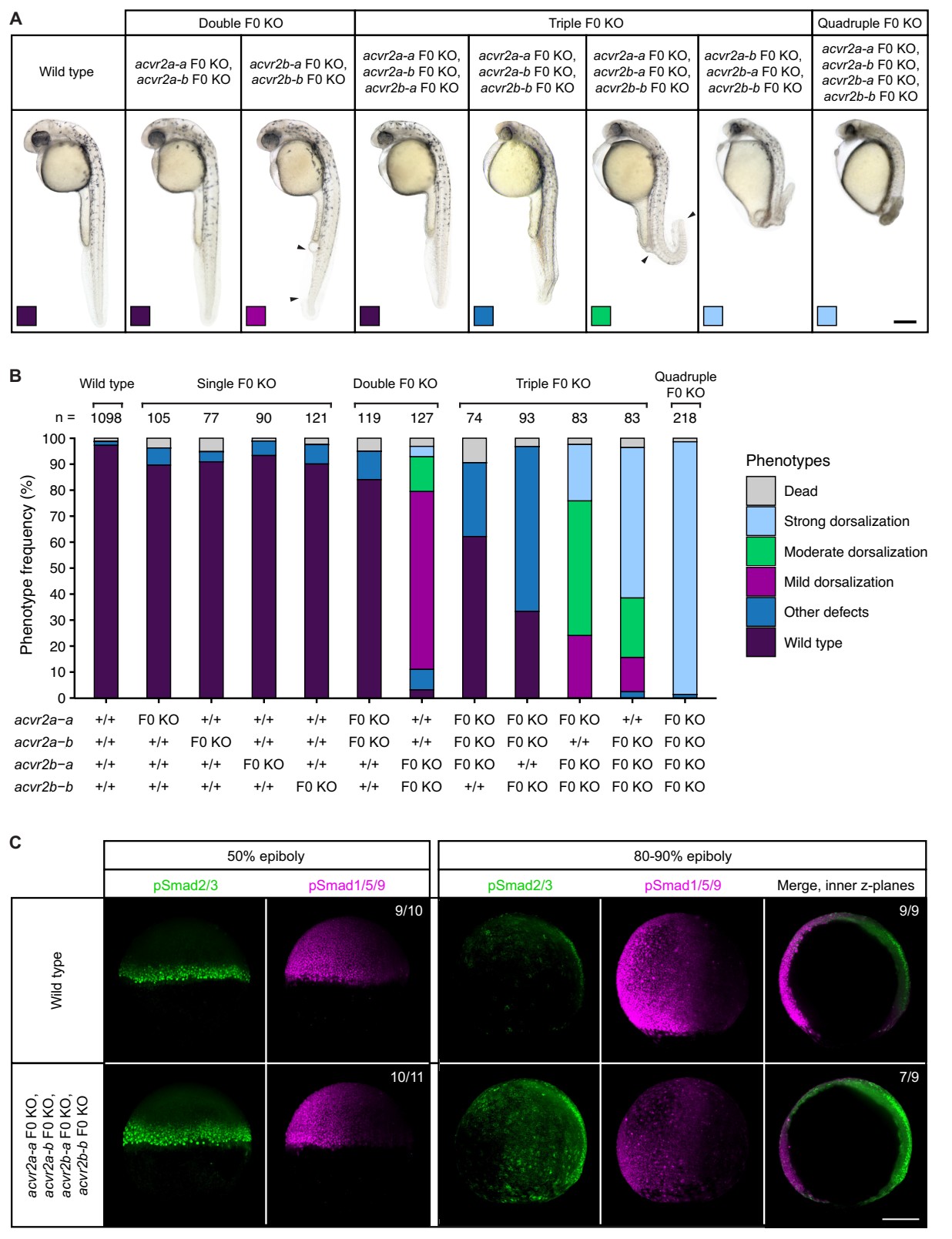

**Figure 2.** Combinatorial removal of putative Type II Nodal receptors causes dorsal-ventral patterning defects. (**A–B**) Phenotypes of embryos upon single, double, triple, and quadruple CRISPR F0 KO of *acvr2* receptors. (**A**) Lateral view of embryos of the indicated condition approximately 28–31 hpf. Arrowheads indicate the extent of ventral fin loss. Boxes indicate the phenotype class according to the scheme presented in (**B**). Scale bar represents 250 μm. (**B**) Frequency of phenotypes observed in embryos of the indicated condition at 1 dpf. n indicates the number of analyzed embryos. (**C**) Nodal

*Figure 2 continued on next page*

*Figure 2 continued*

and BMP signaling visualized by pSmad2/3 and pSmad1/5/9 immunostaining, respectively, in wild-type and quadruple *acvr2* F0 KO embryos at 50% and 80–90% epiboly. Maximum intensity projections show lateral views with dorsal to the right. The number of embryos with the presented phenotype is indicated. Scale bar represents 200 µm. See the *Figure 2—source data 1* file for source data.

The online version of this article includes the following source data and figure supplement(s) for figure 2:

**Source data 1.** Source data for *Figure 2*.

**Figure supplement 1.** Single *acvr2* receptor mutant or knockdown embryos have no obvious patterning defects.

**Figure supplement 1—source data 1.** Source data for *Figure 2—figure supplement 1*.

**Figure supplement 2.** Combinatorial removal of *acvr2* receptors using morpholinos, CRISPR F0 KO and mutants causes axis-patterning defects.

**Figure supplement 2—source data 1.** Source data for *Figure 2—figure supplement 2*.

**Figure supplement 3.** Validation of gRNAs targeting putative Type II Nodal receptors for F0 KO.

**Figure supplement 3—source data 1.** Uncropped pictures of agarose gels showing the T7 Endonuclease I (T7E1) assay results for gRNAs targeting *acvr2a-a* (**A**) and *acvr2a-b* (**B**) presented in *Figure 2—figure supplement 3A, B*.

**Figure supplement 3—source data 2.** Uncropped pictures of agarose gels showing the T7 Endonuclease I (T7E1) assay results for gRNAs targeting *acvr2b-a* (**A**) and *acvr2b-b* (**B**) presented in *Figure 2—figure supplement 3C, D*.

**Figure supplement 3—source data 3.** Raw, unedited picture of the agarose gel showing the T7 Endonuclease I (T7E1) assay results for gRNAs targeting *acvr2a-a* presented in *Figure 2—figure supplement 3A*.

**Figure supplement 3—source data 4.** Raw, unedited picture of the agarose gel showing the T7 Endonuclease I (T7E1) assay results for gRNAs targeting *acvr2a-b* presented in *Figure 2—figure supplement 3B*.

**Figure supplement 3—source data 5.** Raw, unedited picture of the agarose gel showing the T7 Endonuclease I (T7E1) assay results for gRNAs targeting *acvr2b-a* presented in *Figure 2—figure supplement 3C*.

**Figure supplement 3—source data 6.** Raw, unedited picture of the agarose gel showing the T7 Endonuclease I (T7E1) assay results for gRNAs targeting *acvr2b-b* presented in *Figure 2—figure supplement 3D*.

To ensure that the observed phenotypes are specific to the loss of the Acvr2 receptors, we additionally combinatorially disrupted the *acvr2* genes by quadruple morpholino-mediated knockdown and zygotic F0 KO of *acvr2b-b* in the incross progeny of a triple-heterozygous *acvr2a-a*$^{+/-}$;*acvr2a-b*$^{+/-}$;*acvr2b-a*$^{+/-}$ mutant line. The resulting embryos were analyzed for patterning defects at 1 dpf and, in the case of the mutant line, subsequently genotyped. Both approaches reliably recapitulated the phenotypes observed for the quadruple F0 *acvr2* KO (*Figure 2—figure supplement 2A-C*). Furthermore, the mutants confirmed the difference between the *acvr2* genes observed in the combinatorial F0 KO: While *acvr2a-a* and *acvr2a-b*, on their own, were not able to rescue the loss of the other three receptors, the most severe phenotypes were only observed in the absence of functional *acvr2b-a* or *acvr2b-b* alleles (*Figure 2A and B*; *Figure 2—figure supplement 2C*).

To directly determine potential consequences of combinatorial *acvr2* loss on Nodal and BMP signaling, we performed immunostainings for the Nodal signal transducer phosphorylated Smad2/3 (pSmad2/3) and the BMP signal transducer phosphorylated Smad1/5/9 (pSmad1/5/9) in wild-type and quadruple F0 KO embryos at early and late gastrulation stages (50% and 80–90% epiboly, respectively). At 50% epiboly, the pSmad2/3 signal was elevated in quadruple F0 KO embryos, while the pSmad1/5/9 signal appeared unaffected (*Figure 2C*). Towards the end of gastrulation (80–90% epiboly), quadruple F0 KO embryos still showed an increased pSmad2/3 signal compared to wild-type embryos, but in addition a weaker pSmad1/5/9 signal (*Figure 2C*). This indicates that in the absence of the Acvr2 receptors Nodal signaling increases while, with some delay, BMP signaling decreases. To further test whether the Acvr2 receptors act within the same pathway as Nodal, we performed quadruple receptor F0 KO in MZ*oep* mutant embryos and observed their phenotypes at 1 dpf. In addition to MZ*oep*-specific defects, including cyclopia, these embryos also had a shortened body axis and curled tails (*Figure 2—figure supplement 2D*), indicating at least partially additive patterning defects.

Overall, our results show that the zebrafish receptors Acvr2a-a, Acvr2a-b, Acvr2b-a, and Acvr2b-b act partly redundantly and are essential for the formation of the embryonic body plan. However, they do not appear to fulfill the function of classical Type II Nodal receptors. It will be interesting in the future to determine the mechanism by which the Type II receptors mediate partially Nodal-independent patterning processes.

## The Type I receptors Acvr1b-a and Acvr1b-b redundantly mediate Nodal signaling during zebrafish germ layer patterning

We next investigated the role of the putative Type I Nodal receptors Acvr1b-a, Acvr1b-b, and Acvr1c in early zebrafish development. We first examined the morphological phenotypes of single receptor loss of function using mutants, morpholino-mediated knockdown and CRISPR F0 KO as described above. Similar to the Acvr2 receptors, loss of individual Acvr1 receptors did not cause severe phenotypes apart from the non-specific head or tail defects observed for some morpholino treatments (*Figure 3A and C*; *Figure 3—figure supplement 1A-C and E-H*).

To test whether the putative Type I receptors may redundantly mediate Nodal signaling during germ layer patterning, we used combinatorial receptor loss-of-function approaches for *acvr1b-a* and *acvr1b-b*. Morpholino-mediated double knockdown of *acvr1b-a* and *acvr1b-b* resulted in a clear loss of head mesoderm at 1 dpf, leading to the distinctive fused-eye phenotype and a curved body axis associated with loss-of-Nodal signaling (*Figure 3—figure supplement 2A, B*). However, somites still formed in the trunk region, suggesting an incomplete loss of Nodal signaling possibly due to maternal deposition of receptor proteins or incomplete mRNA knockdown. Similar but slightly milder phenotypes were obtained with the double *acvr1b-a* and *acvr1b-b* F0 KO: Most embryos developed two eyes but showed reduced spacing between them (convergent eyes) recapitulating the zygotic F0 KO of *oep* (*Figure 3A and C*), indicating a maternal contribution. Therefore, we injected *acvr1b-b*-targeting morpholinos into maternal-zygotic *acvr1b-a*$^{-/-}$ mutant embryos. This combination reliably recapitulated the full Nodal loss-of-function phenotype at 1 dpf (*Figure 3A and C*). Interestingly, knockdown of *acvr1b-b* in *acvr1b-a*$^{-/-}$ mutants only resulted in Nodal loss-of-function phenotypes when the ATG-targeting morpholino ('MO-1') was used, but not with the splice site-targeting morpholino ('MO-2') (*Figure 3—figure supplement 2C*). Since the splice site-targeting morpholino does not affect the maternally deposited *acvr1b-b* mRNA (i.e. the mRNA at 512-cell stage) (*Figure 3—figure supplement 2D*), this observation indicates that maternally deposited *acvr1b-b* mRNA contributes to proper germ layer formation.

The phenotypes observed in embryos lacking functional *acvr1b-a* and *acvr1b-b* suggest a loss of Nodal signaling. To test this hypothesis, we directly assessed Nodal signaling in these embryos by staining for pSmad2/3 during early gastrulation at 50% epiboly and shield stage (*Figure 3B*; *Figure 4A*). The range of Nodal signaling at shield stage was quantified as the number of pSmad2/3-positive nuclei tiers on the dorsal side (*Figure 4B*) or the normalized intensity of nuclei from the dorsal or lateral side relative to their distance from the margin (*Figure 4—figure supplement 1*). We observed pSmad2/3-positive cells over a distance of about 12-cell tiers at the embryonic margin of wild-type embryos (*Figure 4A and B*) similar to previous reports (*Almuedo-Castillo et al., 2018*; *Lord et al., 2021*; *Rogers et al., 2017*; *van Boxtel et al., 2015*). *acvr1b-a*$^{-/-}$ mutants and embryos injected with *acvr1b-b*-targeting morpholinos had a Nodal signaling range similar to untreated wild-type embryos on the dorsal side. Laterally, *acvr1b-a*$^{-/-}$ mutants showed a slight reduction in pSmad2/3 signal intensity at distances above 30 µm (*Figure 4—figure supplement 1*). In contrast, combined mutation/knockdown of both Type I receptors almost completely abolished the pSmad2/3 signal throughout the embryo (*Figure 4A and B*; *Figure 4—figure supplement 1*). Importantly, the range of pSmad2/3-positive nuclei could be restored to a near-normal extent by injection of 50 pg *acvr1b-a* or 25 pg *acvr1b-b* mRNA (*Figure 4A and B*), and up to 60% of injected embryos displayed normal or partially rescued phenotypes at 1 dpf (*Figure 4C*). These results demonstrate that *acvr1b-a* and *acvr1b-b* redundantly mediate Nodal signaling during germ layer patterning.

## Nodal receptors affect Nodal dispersal in zebrafish embryos

During gastrulation, the establishment of the correct range of Nodal signaling is thought to be crucial for normal germ layer patterning (reviewed in *Rogers and Müller, 2019*). It has previously been hypothesized that the interaction of Nodal with its receptors might control signal propagation (*Müller et al., 2012*), and the strong affinity of Nodals for the receptor Acvr2b-a has been suggested to shape the Nodal gradient (*Wang et al., 2016*). Furthermore, it has recently been shown that the co-receptor Oep can dramatically alter the Nodal signaling range and distribution (*Lord et al., 2021*), but the effect of the receptors on the distribution of Nodal ligands has not been assessed. To test whether Nodal receptors can indeed affect Nodal distribution during germ layer patterning, we transplanted clones expressing Squint-GFP or Cyclops-GFP into the embryonic animal pole (*Soh et al., 2020*;

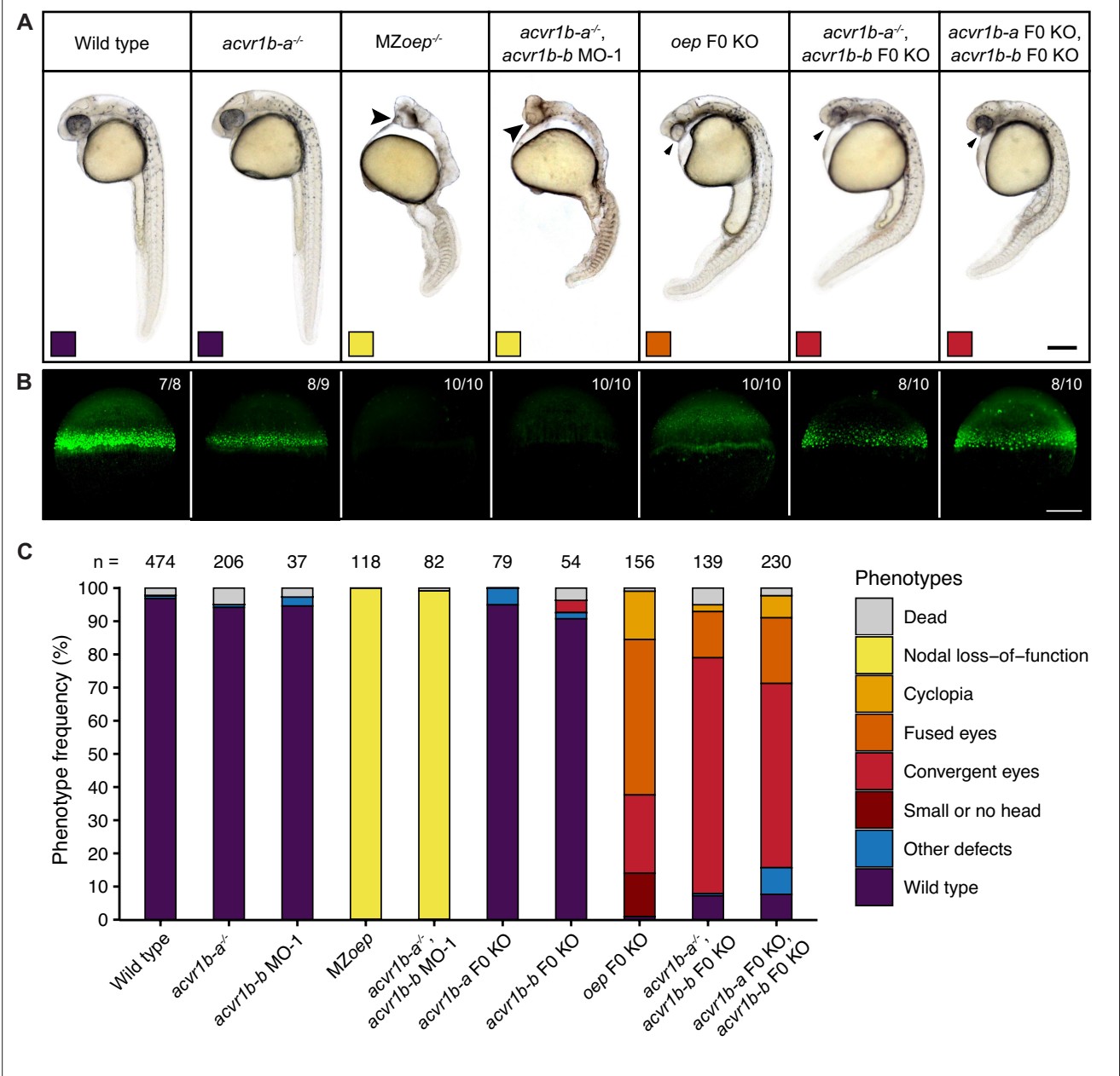

**Figure 3.** Combinatorial removal of putative Type I Nodal receptors causes Nodal-specific patterning defects. Phenotypes of wild-type, MZ*oep* and *oep* CRISPR F0 KO embryos compared to embryos depleted of either or both *acvr1b-a* and *acvr1b-b* using morpholino KDs, CRISPR F0 KOs and mutants. (**A**) Lateral view of embryos of the indicated condition approximately 28–31 hpf. Large arrowheads point to a single cyclopic eye, small arrowheads to fused or convergent eyes. Boxes indicate the phenotype class according to the scheme presented in (**C**). Scale bar represents 250 µm. (**B**) Nodal signaling visualized by pSmad2/3 immunostaining in embryos of the indicated condition (**A**) at 50% epiboly. Maximum intensity projections show lateral views. The number of embryos with the presented phenotype is indicated. Scale bar represents 200 µm. (**C**) Frequency of phenotypes observed in embryos of the indicated condition 1 dpf. n indicates the number of analyzed embryos. Note that one of the gRNAs used for *acvr1b-b* F0 KO has *acvr1b-a* as a predicted off-target, likely explaining the rare occurrences of the convergent eyes phenotype (see *Materials and methods*). See the *Figure 3—source data 1* file for source data.

The online version of this article includes the following source data and figure supplement(s) for figure 3:

**Source data 1.** Source data for *Figure 3*.

**Figure supplement 1.** Single Type I Nodal receptor mutants or knockdown embryos have no obvious patterning defects.

**Figure supplement 1—source data 1.** Source data for *Figure 3—figure supplement 1*.

**Figure supplement 2.** Combinatorial removal of *acvr1b-a* and *acvr1b-b* using morpholinos and mutants mimic Nodal loss-of-function phenotypes.

*Figure 3 continued on next page*

*Figure 3 continued*

**Figure supplement 2—source data 1.** Source data table for *Figure 3—figure supplement 2B, C*.

**Figure supplement 2—source data 2.** Uncropped picture of the agarose gel showing the effect of *acvr1b-b* splice site-targeting morpholino (MO-2) on the maternal and zygotic transcript presented in *Figure 3—figure supplement 2D*.

**Figure supplement 2—source data 3.** Raw, unedited picture of the agarose gel showing the effect of *acvr1b-b* splice site-targeting morpholino (MO-2) on the maternal and zygotic transcript presented in *Figure 3—figure supplement 2D*.

**Figure supplement 3.** Confirmation of gRNAs targeting putative Type I Nodal receptors and *oep* for F0 KO.

**Figure supplement 3—source data 1.** Uncropped pictures of agarose gels showing the T7 Endonuclease I (T7E1) assay results for gRNAs targeting *acvr1b-a* (**A**) and *acvr1b-b* (**B**) presented in *Figure 3—figure supplement 3A, B*.

**Figure supplement 3—source data 2.** Uncropped picture of the agarose gel showing the T7 Endonuclease I (T7E1) assay results for gRNAs targeting *oep* presented in *Figure 3—figure supplement 3C*.

**Figure supplement 3—source data 3.** Raw, unedited picture of the agarose gel showing the T7 Endonuclease I (T7E1) assay results for gRNAs targeting *acvr1b-a* presented in *Figure 3—figure supplement 3A*.

**Figure supplement 3—source data 4.** Raw, unedited picture of the agarose gel showing the T7 Endonuclease I (T7E1) assay results for gRNAs targeting *acvr1b-b* presented in *Figure 3—figure supplement 3B*.

**Figure supplement 3—source data 5.** Raw, unedited picture of the agarose gel showing the T7 Endonuclease I (T7E1) assay results for gRNAs targeting *oep* presented in *Figure 3—figure supplement 3C*.

*Figure 5A*) or mimicked the secretion of endogenous Nodal from the marginal zone by injecting *squint-GFP* or *cyclops-GFP* mRNA (*Müller et al., 2012*) into the yolk syncytial layer (YSL) (*Figure 5—figure supplement 1A*), similar to experiments previously executed for the co-receptor Oep (*Lord et al., 2021*). We then measured the distribution of the tagged Nodal proteins in wild-type embryos and receptor knockout/knockdown conditions (*Figure 5B–D*; *Figure 5—figure supplement 1B-D*).

In wild-type embryos, Squint-GFP and Cyclops-GFP were secreted from the transplanted clones or the YSL and formed graded distributions (*Figure 5B-D*; *Figure 5—figure supplement 1B-D*), similar to previously reported gradients with Squint-GFP being localized relatively diffusely in the extracellular space and Cyclops-GFP being distributed in a punctate pattern (*Müller et al., 2012*; *Soh et al., 2020*; *Wang et al., 2016*). In line with *Lord et al., 2021*, loss-of-function conditions for the co-receptor *oep* led to a broader Squint-GFP distribution (*Figure 5B and C*; *Figure 5—figure supplement 1B,C*). Similarly, disruption of the Type I receptors Acvr1b-a and Acvr1b-b expanded the range of Squint-GFP (*Figure 5B,C*; *Figure 5—figure supplement 1B,C*). Furthermore, Type I receptor or co-receptor loss of function had a drastic effect on the Cyclops-GFP gradient, broadening its range and increasing the number of Cyclops-GFP puncta (*Figure 5B and D*; *Figure 5—figure supplement 1B,D*). Thus, similar to the co-receptor Oep (*Lord et al., 2021*), the Nodal receptors Acvr1b-a/Acvr1b-b affect the dispersal of Nodal proteins in the embryo.

## Receptor binding influences signal propagation through multiple mechanisms

Receptors can affect signal propagation through embryonic tissues by several mechanisms. First, receptor availability can affect the clearance rate of bound ligands and thereby affect signal propagation by modulating protein stability (reviewed in *Rogers and Müller, 2019*; *Rogers and Schier, 2011*). Second, transient receptor binding might slow down signal diffusion (*Crank, 1979*; *Miura et al., 2009*; *Müller et al., 2012*; *Müller et al., 2013*). Third, positive autoregulation through ligand-receptor interactions can extend a ligand's expression domain by relay signaling (*Rogers and Müller, 2019*; *van Boxtel et al., 2015*). To determine whether the receptors affect Nodal propagation by one of these mechanisms, we measured stability and diffusion of Nodal in the presence and absence of receptors and assessed the range of Nodal signaling with and without positive autoregulation.

It has previously been shown that Nodals bind to the Type II receptor Acvr2b-a with nanomolar affinity in living zebrafish embryos (*Wang et al., 2016*). To test whether this interaction affects extracellular ligand stability, we used functional Squint-Dendra2 and Cyclops-Dendra2 in Fluorescence Decay After Photoconversion (FDAP) assays (*Bläßle and Müller, 2015*; *Müller et al., 2012*; *Rogers et al., 2015*). If binding of Nodals to Acvr2b-a affects ligand stability, elevated Acvr2b-a levels should increase the clearance of Squint-Dendra2 and Cyclops-Dendra2. However, overexpression of *acvr2b-a* did not markedly change Nodal clearance rate constants compared to wild-type embryos (*Figure 6A*).

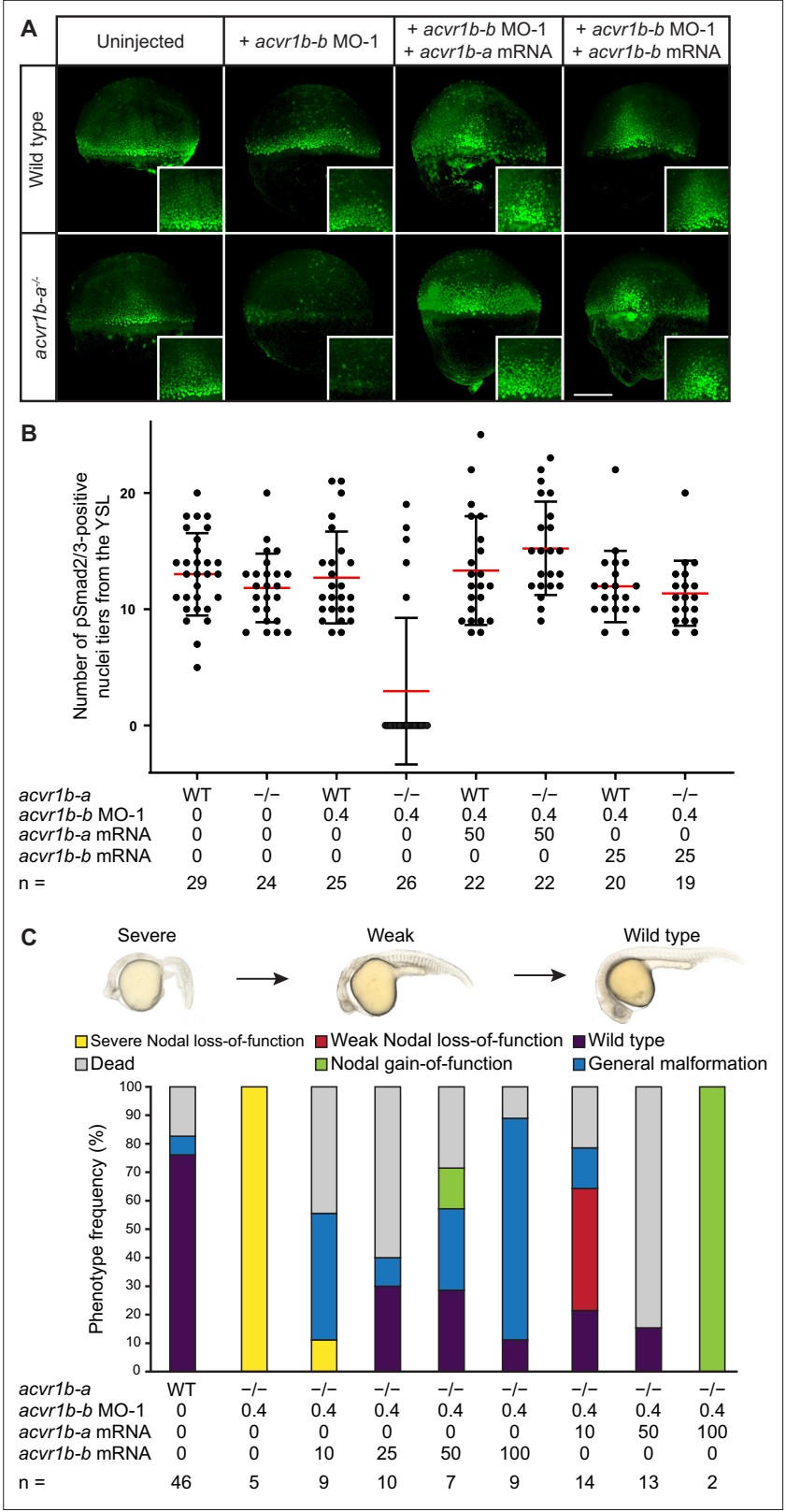

**Figure 4.** The Type I receptors Acvr1b-a and Acvr1b-b redundantly mediate Nodal signaling. (**A,B**) Influence of *acvr1b-a* and *acvr1b-b* on the Nodal signaling range at shield stage. The range of Nodal signaling in shield-stage wild-type, knockdown and rescued embryos was determined by counting the maximum number of nuclei tiers positive for pSmad2/3 immunostaining from the embryonic margin towards the animal pole (i.e.

*Figure 4 continued on next page*

*Figure 4 continued*

the number of pSmad2/3 positive nuclei tiers at the dorsal side). *acvr1b-a*$^{t03pm/t03pm}$ mutants and 0.4 ng *acvr1b-b* transcriptional start site-targeting morpholino (MO-1) were used for receptor loss-of-function conditions. Receptor loss-of-function was rescued with 50 pg of *acvr1b-a* or 25 pg of *acvr1b-b* mRNA. Data was obtained from three independent replicate experiments. (**A**) Maximum intensity projections show dorsal views. Scale bar represents 200 µm. (**B**) n indicates the number of analyzed embryos. Averages are displayed in red, and error bars show standard deviation. (**C**) Rescue of Type I receptor function after combinatorial mutation/knockdown using *acvr1b-a* and *acvr1b-b* mRNA. To deplete the Type I receptors, the *acvr1b-a*$^{t03pm/t03pm}$ mutant was used in combination with 0.4 ng *acvr1b-b* transcriptional start site-targeting morpholino (MO-1). mRNA amounts are given in pg. n indicates the number of analyzed embryos. Note that strong overexpression of Acvr1b-a receptor-encoding mRNA leads to high lethality or tissue aggregates that eventually disintegrate (termed 'Nodal gain-of-function'). See the ***Figure 4—source data 1*** file for source data.

The online version of this article includes the following source data and figure supplement(s) for figure 4:

**Source data 1.** Source data for ***Figure 4***.

**Figure supplement 1.** The Type I receptors Acvr1b-a and Acvr1b-b redundantly mediate Nodal signaling.

**Figure supplement 1—source data 1.** Source data for ***Figure 4—figure supplement 1***.

---

This suggests that the strong interaction between Nodals and Acvr2b-a (***Wang et al., 2016***) is not sufficient to modulate Nodal protein stability. Similarly, the absence of Type I receptors did not cause a decrease in the clearance rate of Nodals (***Figure 6A***).

The diffusion of signals through tissues can be hindered by their transient binding (rapid binding and unbinding) to extracellular molecules (***Bläßle et al., 2018***; ***Kuhn et al., 2022***; ***Mörsdorf and Müller, 2019***; ***Müller et al., 2013***). To test whether receptor interactions can affect Nodal diffusion, we performed Fluorescence Recovery After Photobleaching (FRAP) assays (***Almuedo-Castillo et al., 2018***; ***Bläßle et al., 2018***; ***Müller et al., 2012***; ***Müller et al., 2013***; ***Pomreinke et al., 2017***; ***Soh and Müller, 2018***). We assessed the effective diffusivity of functional Squint-GFP (***Müller et al., 2012***) in wild-type embryos, in embryos overexpressing the Type II receptor *acvr2b-a* or the co-receptor *oep* and in embryos lacking the Type I receptors *acvr1b-a*/*acvr1b-b* (***Figure 6B***). Overexpression of *acvr2b-a* did not markedly change the effective diffusivity of Squint-GFP (***Figure 6B***), suggesting that the strong interaction between Squint and Acvr2b-a previously shown in vivo (***Wang et al., 2016***) is not sufficient to modulate Squint diffusivity. In contrast, *oep* overexpression reduced the effective diffusivity of Squint-GFP from about 2 µm²/s to ~1 µm²/s, consistent with the increased Nodal signaling range and distribution in the absence of *oep* (***Lord et al., 2021***; ***Figure 5C***; ***Figure 5—figure supplement 1C***), the decreased Nodal distribution with overexpressed *oep* (***Lord et al., 2021***) and the increased bound fraction upon *oep* overexpression in single-molecule tracking experiments (***Kuhn et al., 2022***). Strikingly, the effective diffusivity of Squint-GFP in the absence of *acvr1b-a*/*acvr1b-b* increased to >3 µm²/s, consistent with the broader Squint-GFP distribution in the absence of these Type I receptors (***Figure 5C***; ***Figure 5—figure supplement 1C***). Together, our results indicate that Oep, Acvr1b-a and Acvr1b-b serve not only to transduce signaling activity but also to regulate the spatial range of the signal by modulating its diffusion.

The Nodal signaling pathway features strong autoregulatory feedback by inducing the Nodal ligands Squint and Cyclops as well as the co-receptor Oep and the Type I receptor Acvr1b-a (***Figure 1A and E–F***; ***Bennett et al., 2007***; ***Dougan et al., 2003***; ***Dubrulle et al., 2015***; ***Feldman et al., 2002***). However, the role of this positive feedback for the propagation of Nodal signaling is currently unclear (***Lord et al., 2021***; ***Rogers and Müller, 2019***). We found that the feedback-induced co-receptor Oep and the Type I receptors Acvr1b-a and Acvr1b-b act as diffusion regulators of Nodal (***Figure 5***; ***Figure 6B***), implying that the range of Nodal propagation may – paradoxically – be increased in the absence of positive Nodal feedback in surrounding tissues. To test this prediction, we sought to visualize the activity range of endogenous Nodal signals.

Cyclops and Squint have been shown to activate target genes at a distance (***Chen and Schier, 2001***), and the biophysical properties of tagged zebrafish Nodals support their function as short- to mid-range signals (***Müller et al., 2012***). However, these findings are based on ectopic expression assays and the readout of target genes such as *no tail*, whose transcription is also activated by Nodal-induced FGFs and thus does not directly report Nodal activity (***van Boxtel et al., 2015***). It has therefore been debated whether endogenous Nodals act directly at a distance as initially proposed (***Chen***

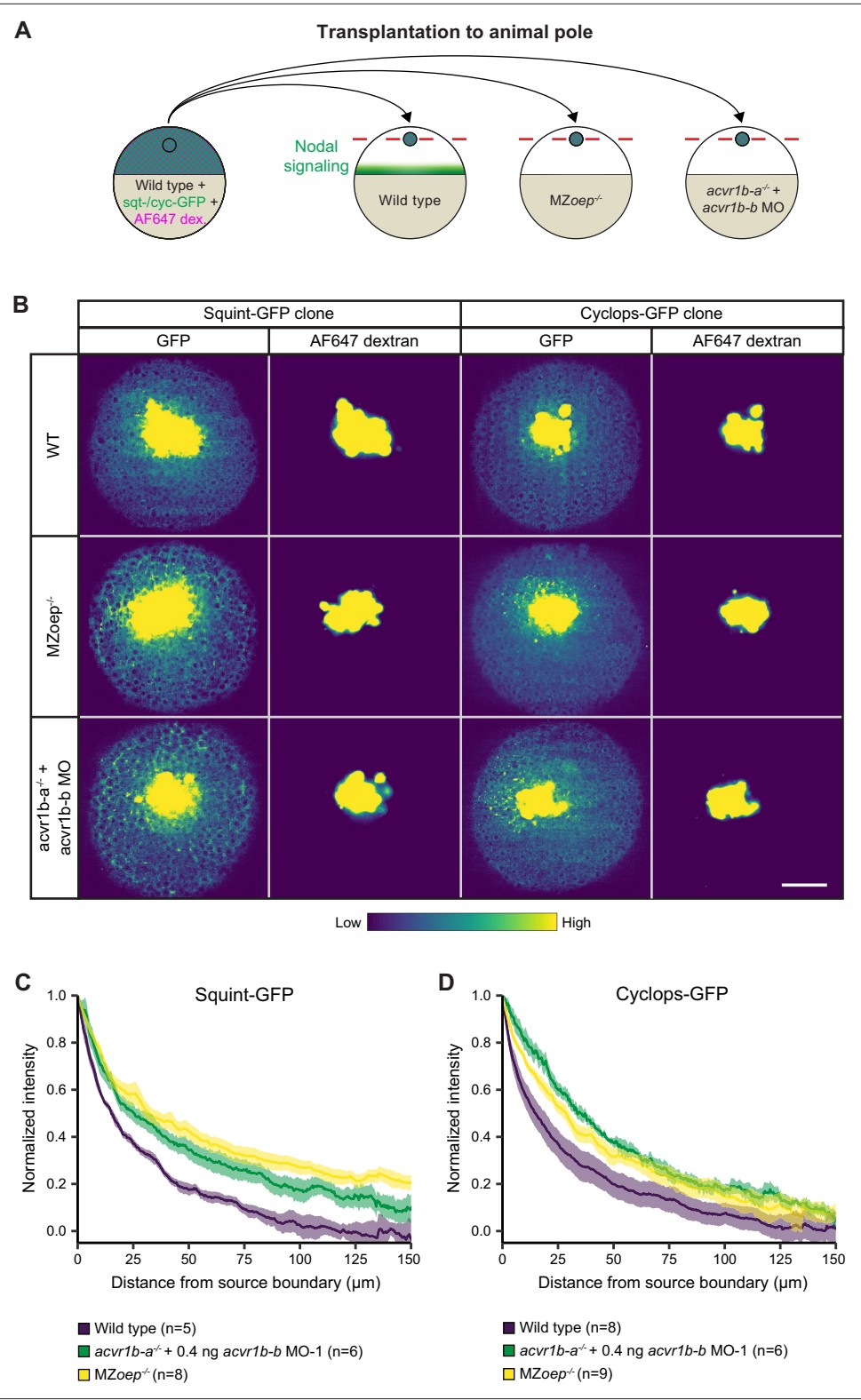

**Figure 5.** Nodal receptors and co-receptors can shape the distribution of Nodal ligands in zebrafish embryos.
(**A**) Schematic of the transplantation assay to create ectopic Nodal signaling sources. Cells from the animal pole of sphere-stage wild-type embryos injected with *squint-GFP* or *cyclops-GFP* mRNA and Alexa Fluor 647 dextran (AF647 dex.) were transplanted to the animal pole of wild-type, MZ*oep*−/−, or acvr1b-a−/− + *acvr1b-b* MO-1 embryos (hosts). Host embryos were imaged 60 min post-transplantation to determine the dispersal of Nodal ligands

*Figure 5 continued on next page*

*Figure 5 continued*

secreted by the clone. (**B**) Animal-pole view of transplanted Squint-GFP or Cyclops-GFP clones in the indicated host embryo 60 min post-transplantation. Single z-slices show the ligand distribution (GFP signal) and the transplanted cells (AF647 dextran signal). Scale bar represents 100 µm. (**C,D**) Quantification of Squint-GFP (**C**) and Cyclops-GFP (**D**) signal distributions in wild-type, MZ*oep*<sup>-/-</sup> and acvr1b-a<sup>-/-</sup> + *acvr1b-b* MO-1 embryos. The mean normalized background-subtracted intensities are shown as a function of their distance from the transplantation site. The error bars (shaded regions) indicate SEM. The number of measured embryos is indicated in parentheses. See the ***Figure 5—source data 1*** file for source data.

The online version of this article includes the following source data and figure supplement(s) for figure 5:

**Source data 1.** Source data for ***Figure 5***.

**Figure supplement 1.** Loss of Acvr1b-a/Acvr1b-b or Oep broadens the range of Squint-GFP and Cyclops-GFP secreted from the YSL.

**Figure supplement 1—source data 1.** Source data for ***Figure 5—figure supplement 1***.

*and Schier, 2001*) or whether they act exclusively at a short range and require relay through positive feedback on Nodal expression (*Liu et al., 2022*; *Rodaway et al., 1999*; *Rogers and Müller, 2019*; *van Boxtel et al., 2015*; *van Boxtel et al., 2018*).

To examine whether untagged zebrafish Nodals can directly act on distant cells at endogenous expression levels and to test the relay model, we transplanted cells from the embryonic margin – where endogenous Nodal expression is highest – of H2A.F/Z:GFP embryos (*Pauls et al., 2001*) into the animal pole – where Nodal expression is absent – of wild-type embryos or Nodal mutant embryos (MZ*sqt*<sup>-/-</sup>;*cyc*<sup>-/-</sup>; ***Figure 6C***). Since MZ*sqt*<sup>-/-</sup>;*cyc*<sup>-/-</sup> embryos cannot produce functional Nodals, there is no Nodal relay in this mutant background allowing us to directly assess the endogenous Nodal signaling range in the absence of Nodal autoinduction or confounding relay effects. To assess Nodal signaling after transplantation, pSmad2/3 immunofluorescence staining was performed on embryos fixed 2 hr post-transplantation. pSmad2/3 can clearly be detected in the nuclei of cells outside the transplant in both wild-type and mutant backgrounds (***Figure 6C***). This indicates that Nodals do not require a relay mechanism to signal to distant cells. Interestingly, the pSmad2/3 intensities inside and around the transplants were higher in the MZ*sqt*<sup>-/-</sup>;*cyc*<sup>-/-</sup> background than in the wild-type background (***Figure 6—figure supplement 2A,D***), and pSmad2/3-positive nuclei were found more frequently outside of transplanted clones in the MZ*sqt*<sup>-/-</sup>;*cyc*<sup>-/-</sup> background (***Figure 6C***), consistent with our prediction that the range of Nodal propagation should be increased in the absence of positive Nodal feedback in tissues surrounding the Nodal source. However, in the absence of Nodal signaling Leftys are not expressed (*Feldman et al., 2002*; *van Boxtel et al., 2015*), which might also contribute to the extended Nodal signaling range in MZ*sqt*<sup>-/-</sup>;*cyc*<sup>-/-</sup> mutants.

Endogenous Nodal signaling is active at the embryonic margin, and we therefore wanted to assess whether Nodals can also signal over a distance in marginal tissues, where the feedback-regulated receptors Oep and Acvr1b-a are expressed (***Figure 1D–F***; *Vopalensky et al., 2018*). We therefore performed margin-to-margin transplantations of H2A.F/Z:GFP cells into MZ*sqt*<sup>-/-</sup>;*cyc*<sup>-/-</sup> host embryos and found that Nodals can also act on distant cells at the margin (***Figure 6D***), where receptor expression is higher than in the animal pole (***Figure 1D and F***). In agreement with our prediction that the range of Nodal propagation should be increased with dampened positive feedback in tissues surrounding the Nodal source, MZ*sqt*<sup>-/-</sup>;*cyc*<sup>-/-</sup> mutant embryos showed increased pSmad2/3 intensities and pSmad2/3-positive nuclei tended to be found more frequently outside of transplants compared to wild-type hosts, with a few cases even showing extremely extended ranges (***Figure 6D***; ***Figure 6—figure supplement 2B,E***). While our findings are consistent with the idea that positive feedback mediated by receptors and co-receptors restricts the range of Nodal signaling, we cannot rule out that the extended range we observed with margin transplantations in MZ*sqt*<sup>-/-</sup>;*cyc*<sup>-/-</sup> hosts is also influenced by dampened negative feedback, which will have to be tested in MZ*sqt*<sup>-/-</sup>;*cyc*<sup>-/-</sup>;*lft1*<sup>-/-</sup>;*lft2*<sup>-/-</sup> quadruple mutants in the future.

## Discussion

The Nodal signaling pathway is a key regulator of vertebrate development and is important for human disease and regenerative medicine (*Lee et al., 2010*; *Roessler et al., 2008*; *Schier, 2009*; *Tewary*

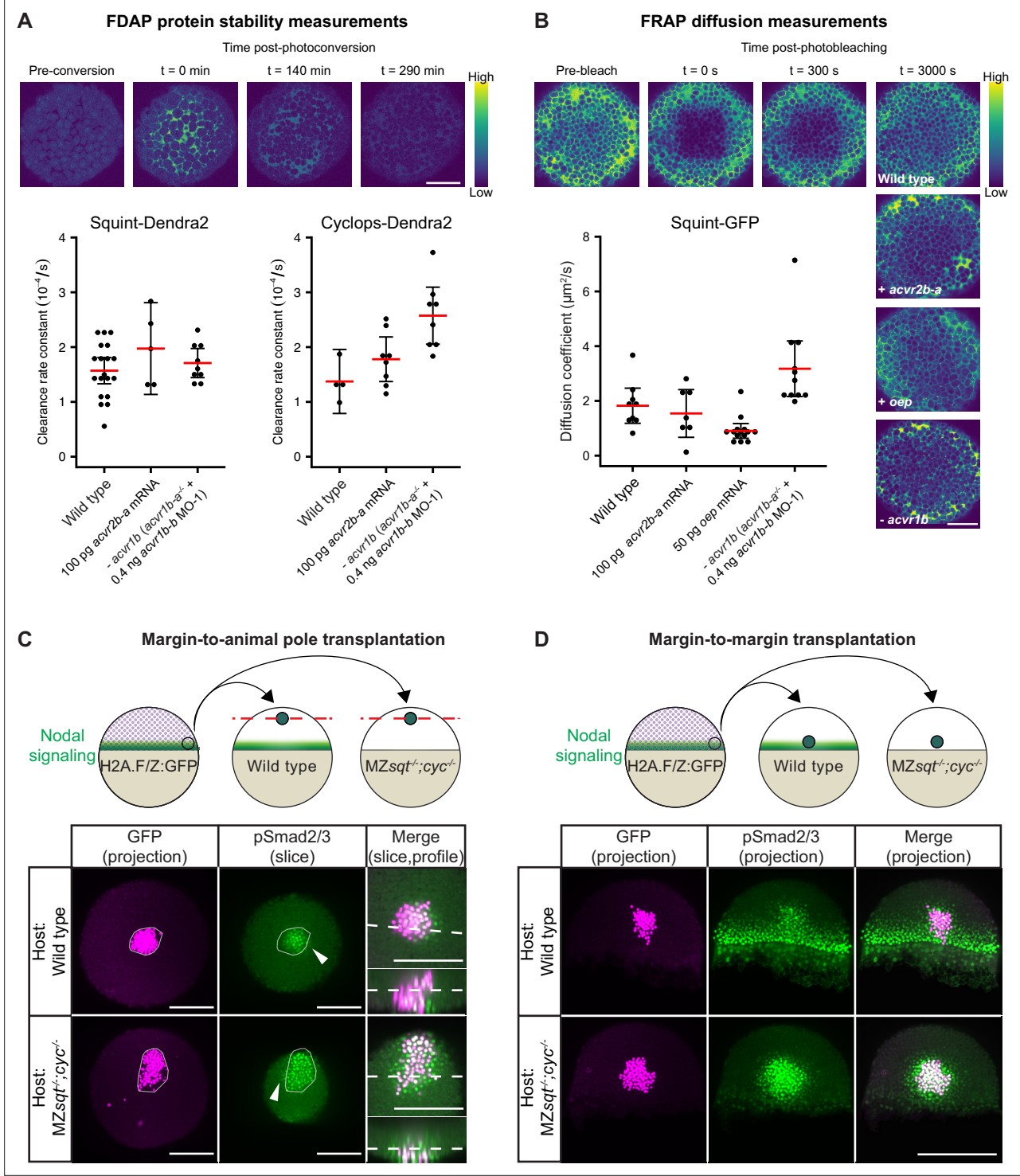

**Figure 6.** Influence of Nodal receptors on Nodal stability, diffusivity and autoregulatory signal propagation. (**A**) Impact of *acvr1* loss-of-function and *acvr2b-a* overexpression on Squint- and Cyclops-Dendra2 clearance rate constants determined using FDAP measurements. For *acvr1* loss-of-function, 0.4 ng *acvr1b-b* MO-1 were injected into *acvr1b-a^-/-* mutant embryos. For overexpression, 100 pg *acvr2b-a* mRNA were injected into wild-type embryos. Mean extracellular clearance rate constants are displayed in red, and individual measurements are shown as black dots. Error bars represent 95% confidence intervals. See **Figure 6—figure supplement 1A** for representative fits. (**B**) Influence of receptor levels on Squint- and Cyclops-GFP diffusivities determined using FRAP measurements. For overexpression, either 50 pg *oep* mRNA or 100 pg *acvr2b-a* mRNA were injected into wild-type embryos at the one-cell stage. *acvr1b-a* mutants and 0.4 ng *acvr1b-b* transcriptional start site-targeting morpholino (MO-1) were used for receptor loss-of-function conditions. The mean diffusion coefficients are displayed in red, and individual measurements are shown as black dots. Error bars represent

*Figure 6 continued on next page*

*Figure 6 continued*

95% confidence intervals. See *Figure 6—figure supplement 1B* for representative fits. Scale bars represents 100 μm. (**C**) Margin-to-animal pole transplantations show that Nodals at endogenous expression levels can signal to distant cells. Top panel: Experimental setup of the margin-to-animal pole transplantations, in which wild-type embryos or MZ*sqt$^{-/-}$;cyc$^{-/-}$* embryos that lack Nodal relay were used as hosts. Bottom panel: Immunofluorescent stainings show that pSmad2/3-positive nuclei (green) are detected outside of the transplanted clones (magenta) in both wild-type (top row) and MZ*sqt$^{-/-}$;cyc$^{-/-}$* (bottom row) hosts. (**D**) Margin-to-margin transplants show that Nodals at endogenous expression levels can signal to distant cells at the embryonic margin. Top panel: Experimental setup. Bottom panel: Representative maximum intensity projections of immunofluorescent stainings. Transplantations into wild-type embryos (top row) and MZ*sqt$^{-/-}$;cyc$^{-/-}$* embryos (bottom row) are shown. Scale bars represent 200 μm. Animal pole views are shown in (**A–D**). See the *Figure 6—source data 1* file for source data and sample size.

The online version of this article includes the following source data and figure supplement(s) for figure 6:

**Source data 1.** Source data for *Figure 6*.

**Figure supplement 1.** Quantification of Nodal stability and diffusivity in receptor mutant or receptor overexpression embryos.

**Figure supplement 1—source data 1.** Source data for *Figure 6—figure supplement 1*.

**Figure supplement 2.** Increased pSmad2/3 signal intensities and ranges from transplants in Nodal-mutant hosts compared to wild-type hosts.

**Figure supplement 2—source data 1.** Source data for *Figure 6—figure supplement 2*.

*et al., 2019*). Here, we systematically assessed putative Nodal Type I and Type II receptor homologs in zebrafish. We found that the transcripts of most of these putative Nodal receptors are maternally deposited and present during germ layer patterning, indicative of a potential role in early patterning. The Type I receptor Acvr1c (Alk7) is an exception and not expressed until 4 dpf, making it unlikely to be involved in germ layer formation. While single mutants of the Nodal co-receptor Oep and the signal transducer Smad2 display complete loss-of-function phenotypes (*Dubrulle et al., 2015*; *Gritsman et al., 1999*), the loss of individual Nodal ligands (Squint and Cyclops) only leads to partial defects (*Dougan et al., 2003*; *Feldman et al., 1998*; *Rebagliati et al., 1998a*; *Rebagliati et al., 1998b*; *Sampath et al., 1998*). This redundancy is mirrored in the function of the Type I and Type II receptors. For example, individual loss of *acvr1b-a* and *acvr1b-b* activity did not induce Nodal-related defects, whereas combined loss-of-function conditions for both *acvr1b-a* and *acvr1b-b* led to a complete Nodal mutant phenotype, suggesting that these Type I receptors redundantly mediate Nodal signaling during early embryogenesis.

Similar to the Type I receptors, only combinatorial loss of the putative Type II receptors Acvr2a-a, Acvr2a-b, Acvr2b-a, and Acvr2b-b caused embryonic patterning defects. In contrast to the Type I receptors, however, the loss of the Acvr2 receptors did not phenocopy Nodal loss-of-function phenotypes, caused an elevation rather than a reduction of pSmad2/3, and, in MZ*oep* embryos, led to patterning defects independent of Nodal (*Figure 2*; *Figure 2—figure supplement 2*). The phenotype observed upon quadruple *acvr2* receptor KO partly resembles the mutant phenotype of *pgy$^{dty40}$* (*Mullins et al., 1996*), a hypomorphic mutation of the *smad5* gene (also called *somitabun*) with a weak antimorphic effect (*Kramer et al., 2002*), which could indicate an influence of the receptors on the BMP signaling pathway. In line with this hypothesis, quadruple *acvr2* KO embryos showed a reduction in pSmad1/5/9 signal (*Figure 2C*). However, this reduction in BMP signaling was only visible at late gastrulation, a few hours after the embryos displayed an increase in pSmad2/3. Moreover, some of the morphological defects, most obviously the reduction in head and eye size, deviated from the classical BMP loss-of-function phenotype. It is currently unknown how the loss of Acvr2 receptors causes these effects and through what mechanisms and signals the receptors mediate embryonic patterning. It is tempting to speculate that receptor promiscuity might play a role. Acvr2b-a, for example, has been shown to be able to mediate both Activin and BMP signaling by recruiting the respective Type I receptor Acvr1b-a or Bmpr1a (*Nagaso et al., 1999*). Even a direct high-affinity interaction of Nodal with the BMP Type II receptor Bmpr2 has been shown in vitro (*Aykul et al., 2015*). Furthermore the Type I receptor TGFβr1 can phosphorylate and thereby activate the Type I receptor Acvr1, indicating that Type I receptors can function like Type II receptors under certain conditions (*Ramachandran et al., 2018*). Whether this dual function of Type I receptors affects endogenous Nodal signaling in zebrafish requires further investigation.

During germ layer patterning, Nodal is first expressed in the YSL, from which it spreads into the embryo to form a signaling gradient. There are currently two major models that can explain the propagation of Nodal signaling in this context. In the hindered diffusion model, Nodal is secreted

from source cells and its free diffusion through the embryo is hindered by interactions with immobile diffusion regulators (*Müller et al., 2012*; *Müller et al., 2013*; *Rogers and Müller, 2019*; *Wang et al., 2016*; *Lord et al., 2021*; *Kuhn et al., 2022*). In the second model, Nodal ligands only act in a juxtacrine fashion, and propagation of Nodal signaling to adjacent cells is mediated by a relay mechanism involving positive feedback of Nodal expression (*Liu et al., 2022*; *Lord et al., 2021*; *Rogers and Müller, 2019*; *van Boxtel et al., 2015*; *van Boxtel et al., 2018*). To distinguish between these models, we transplanted cells expressing endogenous Nodal signals into Nodal-mutant backgrounds that are devoid of relay mechanisms involving feedback on Nodal expression. Consistent with the known function of Nodals as short- to mid-range signals (*Chen and Schier, 2001*; *Müller et al., 2012*), we found that Nodals do not exclusively act in a juxtacrine manner and can signal to distant cells even in the absence of Nodal relay (*Figure 6C and D*). The importance of positive Nodal feedback as an additional mechanism to regulate Nodal signaling propagation is supported by the restriction of the Type I receptor Acvr1b-a and the co-receptor Oep to the marginal zone. This spatial restriction is mediated by Nodal ligands, which are also expressed at the margin (*Bennett et al., 2007*; *Dubrulle et al., 2015*; *Feldman et al., 2002*; *Meno et al., 1999*; *Vopalensky et al., 2018*), suggesting a role for positive feedback to limit Nodal signaling to the embryonic margin. Although our data support the idea that Nodals function as classical morphogens and act directly at a distance as master regulators of mesendoderm formation (*Chen and Schier, 2001*), complex germ layer patterning requires the interaction with other signaling molecules such as FGFs (*Bennett et al., 2007*; *Dubrulle et al., 2015*; *van Boxtel et al., 2015*; *van Boxtel et al., 2018*), which act as secondary downstream relay factors to induce mesendodermal gene expression at the correct time and place.

The action range of Nodals has been proposed to be restricted by extracellular interactions (*Müller et al., 2012*; *Müller et al., 2013*). In this hindered diffusion model, Nodal's free diffusivity of approximately 40 µm$^2$/s (*Müller et al., 2013*; *Wang et al., 2016*) would be slowed down by an order of magnitude through interactions with immobile diffusion regulators such as receptors (*Wang et al., 2016*) or heparin sulfate proteoglycans (*Marjoram and Wright, 2011*). Indeed, our recent single-molecule tracking experiments showed that individual Nodal molecules have binding times of tens of seconds in the extracellular space, leading to hindered diffusion on the nanometer to micrometer scale (*Kuhn et al., 2022*). Here, we directly assessed the influence of Nodal receptors on the dispersal of Nodal ligands at the tissue scale. We found that embryos with reduced receptor levels displayed broader Nodal gradients (*Figure 5B–D*; *Figure 5—figure supplement 1B-D*). To elucidate whether the Type I receptors Acvr1b-a and Acvr1b-b as well as the co-receptor Oep directly regulate Nodal diffusivity, we used FRAP assays to measure Nodal mobility in intact embryos with modulated receptor levels. Consistent with our gradient analyses, receptor and co-receptor had a large impact on Nodal diffusivity (Cohen's $d$=1.03 and 1.4 with p=0.028 and 0.003, respectively; see *Materials and methods*), indicating their importance not only for Nodal signaling, but also as a diffusion regulators during early embryogenesis. In agreement with our findings, it has recently been shown that the Nodal co-receptor Oep restricts Nodal's distribution and signaling (*Lord et al., 2021*) and can locally hinder Nodal diffusion by transient membrane trapping during zebrafish embryogenesis (*Kuhn et al., 2022*). Previous research has demonstrated that Oep is critical for Nodal signaling (*Gritsman et al., 1999*) and indicated that Oep mediates the interaction of Nodal with the Activin receptor complex (*Bianco et al., 2002*; *Reissmann et al., 2001*; *Yan et al., 2002*; *Yeo and Whitman, 2001*). Moreover, the Oep-Nodal interaction is crucial for the regulation of Nodal signaling, as chimeric Nodals that do not require Oep for signaling activity cannot be inhibited by Lefty (*Cheng et al., 2004*). In mouse models, Nodal has been shown to directly interact with the Oep homolog Cripto already before secretion and processing of the Nodal protein, and Oep was also found to regulate Nodal endocytosis and subsequent signaling (*Blanchet et al., 2008a*; *Blanchet et al., 2008b*). While the influence of Oep/Cripto and the Type II receptors on Nodal propagation could be explained by their direct interaction, the Type I receptor Acvr1b-a is thought to require the presence of the co-receptor Oep/Cripto to interact with Nodal (*Reissmann et al., 2001*). However, there is evidence that Nodal can also directly interact with Type I receptors (*Calvanese et al., 2015*; *Reissmann et al., 2001*). Alternatively, the observed impact of Type I receptor levels on Nodal dispersal might be due to a failure in assembling the full Nodal receptor complex, possibly affecting endocytosis of Nodal (*Zhou et al., 2004*) and causing Nodal to accumulate in the extracellular space resulting in a broader Nodal gradient (*Bennett et al., 2007*; *Dubrulle et al., 2015*; *Feldman et al., 2002*; *Meno et al., 1999*; *Vopalensky et al., 2018*).

In summary, we performed a systematic analysis of putative zebrafish Nodal receptors and found that the Type I receptors Acvr1b-a and Acvr1b-b as well as the co-receptor Oep can shape Nodal gradients during early embryogenesis by modulating ligand mobility and dispersal. In the future, it will be interesting to determine the function of receptor redundancy in Nodal signaling, to analyze the role of receptor and co-receptor feedback in robust embryogenesis (*Stapornwongkul et al., 2020*; *Zhu et al., 2020*) and to elucidate the mechanism through which the Acvr2 receptors pattern the early embryo.

## Materials and methods

### Fish lines and husbandry

All procedures were executed in accordance with the guidelines of the State of Baden-Württemberg and approved by the Regierungspräsidium Tübingen and the Regierungspräsidium Freiburg. MZ*oep*^tz57 embryos were generated as previously described (*Gritsman et al., 1999*; *Zhang et al., 1998*). The wild-type strain Tü was used for the generation of the *acvr1c*^t06pm mutant allele. For the generation of *acvr1b-a*^t03pm and *acvr2b-a*^t08pm mutants, the wild-type strain TE was used. The generated alleles contain indels leading to frameshifts resulting in premature stop codons within the first exons: a 4 bp deletion for *acvr1b-a*, a 2 bp deletion for *acvr1c* and a 4 bp deletion for *acvr2b-a* (*Figure 3—figure supplement 1B,C*; *Figure 2—figure supplement 1D*). *acvr2a-a*^sa34654 and *acvr2a-b*^sa18285 mutants were obtained from the European Zebrafish Research Center (EZRC). These mutants carry single-nucleotide mutations leading to alternative splicing and a premature stop codon, respectively (*Figure 2—figure supplement 1B,C*). Unless otherwise stated, maternal-zygotic receptor mutant embryos were used. H2A.F/Z:GFP embryos were obtained from an incross of GFP-positive H2A.F/Z:GFP fish (*Pauls et al., 2001*). MZ*sqt*^-/-;*cyc*^-/- embryos were obtained from an incross of *sqt*^-/-;*cyc*^-/- mutants (*Feldman et al., 1998*; *Schier et al., 1996*) generated by germline transplantation (*Ciruna et al., 2002*). The fish strain TE was used as a wild-type control in all experiments.

### Phylogenetic analysis

For phylogenetic analysis, human and mouse protein sequences of the Type I receptors Acvr1b and Acvr1c as well as protein sequences of the Type II receptors Acvr2a and Acvr2b were used for BLAST queries in Uniprot (RRID: SCR_002380) to identify zebrafish homologs. The alignment of human, mouse and zebrafish sequences was performed using Clustal Omega (*Madeira et al., 2019*). The phylogenetic tree was calculated with a neighbor-joining algorithm using the blosum62 matrix. Jalview version 2.10.3b1 was used for visualization (*Waterhouse et al., 2009*). Branch lengths indicate evolutionary distance.

### 5'RACE

To identify the start of *acvr2a-b*, 5'RACE (Rapid Amplification of cDNA Ends) was conducted. Total RNA was extracted from ten wild-type embryos at shield stage as described below. 5'RACE was performed using the 5'/3' RACE Kit, 2nd Generation (Roche 03353621001) according to the manufacturer's instruction. 1 µg of RNA was used as starting material, and the following gene-specific primers were used for cDNA synthesis (SP1) and PCR amplification (SP2), respectively:

| Primer | Binding site | Sequence (5'->3') |
| --- | --- | --- |
| *acvr2a-b* SP1 | exon 3 (previously annotated as exon 2) | GCTCAACCGTCTCTGGATTG |
| *acvr2a-b* SP2 | exon 3 (previously annotated as exon 2) | ACAAGTTTCCCTCGCAGCAG |

The resulting PCR products were sub-cloned using the Zero Blunt TOPO PCR Cloning Kit (Invitrogen) and analyzed by Sanger sequencing. The resulting sequences contained part of exon 3, the entire exon 2 (previously annotated as exon 2 and 1, respectively) and 143–450 bp upstream of the previously predicted transcriptional start of *acvr2a-b*'s mRNA. Using BLAT search (*Kent, 2002*), the upstream sequence was mapped to chrUn_KN150226v1 with 100% sequence identity. While the exact length of *acvr2a-b*'s 5' UTR varied by approximately 300 bp, all sequenced clones (9/9) supported the same transcriptional start of *acvr2a-b*'s coding sequence (*Figure 1—figure supplement 1A*).

The gene thus contains an additional exon (the new exon 1) at chrUn_KN150226v1:12292–12345 (+strand), which codes for most of the receptor's signal peptide.

## Whole-mount in situ hybridization

To synthesize *acvr1b-a*, *acvr1b-b*, *acvr1c*, *acvr2a-a*, *acvr2a-b*, *acvr2b-a*, and *acvr2b-b* probes for in situ hybridization assays, full-length receptor-encoding sequences amplified from shield stage cDNA were cloned into TOPO Blunt plasmids (Thermo Fisher Scientific 45024) using the following primers:

| Receptor | Forward primer (5'->3') | Reverse primer (5'->3') |
|---|---|---|
| *acvr1b-a* | ATGCTAAGAGATGGGAATGTTGC | TCAGATCTTAATGTCTTCTTGGACG |
| *acvr1b-b* | ATGGACCCACGGCAAATC | TCAGATTTTGAGATCCTCGT |
| *acvr1c* | ATGTCTCATCCCAGGTGCTCAG | TTCTTTAACATCCTTGACCACAGTCAC |
| *acvr2a-a* | ATGGGACCTGCAACAAAGCT | TCATAGACTAGACTCCTTTG |
| *acvr2a-b* | ATGGCGAGCCACTGGACAAACT | TCATAGGCTGGACTCTTTAG |
| *acvr2b-a* | ATGTTCGCTTCTCTGCTCACTTT | TCAGATGCTGGACTCTTTGGGC |
| *acvr2b-b* | ATGTTTGTTCCCTGGCTGGC | TCAGGTGCTGGAGTCTTTGG |

For in situ probe synthesis, plasmids were linearized using SpeI or NotI restriction enzymes followed by in vitro transcription using SP6 or T7 polymerase (Roche) and digoxigenin (DIG)-modified ribonucleotides (Roche). RNA probes were purified using the RNeasy MinElute Cleanup kit (Qiagen 74204) according to the manufacturer's protocol. Embryos fixed in 4% formaldehyde and transferred into methanol for storage were processed for in situ staining as previously described (*Thisse and Thisse, 2008*), but without proteinase K treatment and pre-absorption of the anti-DIG antibody (Sigma-Aldrich, Roche 11093274910).

## mRNA synthesis

Full-length receptor-encoding sequences were amplified from cDNA of shield-stage wild-type TE embryos using the primers listed in the section *Whole-mount in situ hybridization*. The sequences were then re-amplified and cloned into pCS2 +vectors using the following primers and restriction enzyme (RE, obtained from NEB) combinations:

| Receptor | Forward primer (5'->3') | Reverse primer (5'->3') | RE |
|---|---|---|---|
| *acvr1b-a* | TCCCATCGATGCCACCATGC TAAGAGATGGGAATGTTGC | AGAGGCCTTGAATTCGATCAG ATCTTAATGTCTTCTTGGACG | ClaI EcoRI |
| *acvr1b-b* | GATTCGAATTCGCCACCATG GACCCACGGCAAATC | AGAGGCTCGAGCCTTCAGATT TTGAGATCCTCGTCCA | EcoRI XhoI |
| *acvr2b-a* | AGGATCCCATCGATGCCACC ATGTTCGCTT | CACTATAGTTCTAGATCAGAT GCTGGACTCTT | ClaI XbaI |

Plasmids encoding Squint-GFP, Squint-Dendra2, Cyclops-GFP, Cyclops-Dendra, Lefty2-Dendra2 and Oep have been described before (*Gritsman et al., 1999*; *Müller et al., 2012*; *Zhang et al., 1998*). For mRNA synthesis, plasmids were linearized with NotI-HF (NEB R3189). mRNA was generated using the mMessage mMachine SP6 Transcription Kit (Thermo Fisher Scientific AM1340). Synthesized mRNA was purified with RNeasy Mini kits (Qiagen 74104) and dissolved in nuclease-free water.

## Microinjections and embryo dechorionation

For mRNA, sgRNA and morpholino injections, embryos were injected at the one- or two-cell stage with the indicated amounts in a total of 1 nl or 2 nl. For CRISPR F0 KO, 1 nl or 2 nl RNP mixes (see *CRISPR F0 KO*) were injected into the yolk of early one-cell stage embryos. YSL injections were performed by injecting 2 nl of an injection mix containing 100 pg of *squint-GFP* or *cyclops-GFP* mRNA and 0.5 ng of Alexa Fluor 647 dextran (Invitrogen D22914) into the YSL of sphere-stage embryos. Imaging of YSL-injected embryos was started 2 hr post-injection (hpi) for *squint-GFP* injections and 4 hpi for *cyclops-GFP* injections. Ectopic Nodal signaling sources were generated by injecting 250 pg *squint-GFP* or *cyclops-GFP* mRNA and 100 pg of Alexa Fluor 647 dextran (Invitrogen D22914) in a

volume of 2 nl into the cell of one-cell stage wild-type embryos. In each case, injected embryos were incubated at 28 °C, and unfertilized embryos were discarded at 4–5 hpf.

For fixation, imaging, transplantation and YSL injections, embryos were dechorionated manually using forceps or enzymatically using 0.1 mg/ml Pronase (Roche 11459643001) in 5 ml embryo medium (*Rogers et al., 2015*).

*acvr1b-a*, *acvr1c* and *acvr2b-a* mutants were generated using the CRISPR/Cas9 system (*Gagnon et al., 2014*). Target sequences for guide RNAs were chosen using CHOPCHOP (*Montague et al., 2014*). sgRNAs targeting *acvr1b-a* (a mix of sgRNAs targeting GCTACAGCAGTTCGTCGAGG and GGATTACTAGCGGTCGGCGA) and *acvr1c* (AGCGCTGCATCTGAGCACCT) were synthesized as described previously (*Gagnon et al., 2014*). *acvr2b-a* sgRNA (targeting GTTCGCTTCTCTGCTCACTT ) was procured from IDT. A total of 400 pg of Cas9-encoding mRNA (Addgene MLM3613) and 150 pg of sgRNA were co-injected into one- to two-cell-stage wild-type embryos.

## Genotyping

Genomic DNA was isolated from caudal fin tissue of adult zebrafish using the 'hotshot' method (*Meeker et al., 2007*), and regions of interest were amplified using standard PCR conditions and the following primers:

| Mutant | Target | Forward primer (5'->3') | Reverse primer (5'->3') |
|---|---|---|---|
| *acvr1b-a^t03pm* | Exon 2 | TCGCTTGTCAATATCACACACA | CTCTCTCTCCACACACCATCAG |
| *acvr1c^t06pm* | Exon 1 | TCTGTCTACGTGTTGTCGCTTT | AAAGTTGGTGTGTGCTGACAGT |
| *acvr2a-a^sa34654* | Exon 2 | AACTACAACCCCAGCTTGGAGAA | TTTGAAAATTCTTTGAAATCTTT |
| *acvr2a-b^sa18285* | Exon 2 (previously annotated as exon1) | TTTCCAGTTGTGTTTGATTCCATGT | ACAAGTTTCCCTCGCAGCAG |
| *acvr2b-a^t08pm* | Exon 1 | GTGGTGTGTGAGAGTGTGTGTG | CAGGAGCATTTTAACAACACGA |

PCR amplicons were prepared for direct use in sequencing reactions by treatment with ExoI (NEB M0568) and rSAP (NEB M0371L), and the respective amplification primers were used in separate sequencing reactions. Mutations in the first generation were identified using PolyPeakParser (*Hill et al., 2014*) and Hetindel (RRID:SCR_018922). Lasergene Seqman Pro 14 was used for subsequent genotyping analysis. Mutants were outcrossed to wild-type TE fish at least once before incrossing heterozygotes to obtain homozygous fish.

## Morpholino antisense oligonucleotides

For each receptor, several morpholinos targeting splice sites or the region surrounding the ATG start codon were designed. The following morpholinos (ATG start site targets underlined) were obtained from GeneTools (Philomath, OR):

| Target | Morpholino sequence (5'->3') | Target site | Reference |
|---|---|---|---|
| *acvr1b-a 1* | CTGCAACATTCC<u>CAT</u>CTCTTAGCAT | ATG start site | *Jazwińska et al., 2007* |
| *acvr1b-a 2* | GTTTGGCCTGTACTGCTACCATTG | e2i2 splice site | |
| *acvr1b-a 3* | ATAAACATGCAACTTACCAGACCCT | e3i3 splice site | |
| *acvr1b-b 1* | <u>CAT</u>CCTTACAGGACTCCCATTGCAC | ATG start site | |
| *acvr1b-b 2* | CAAAGATTTGTTTTCAGCACCTCCA | e7i7 splice site | |
| *acvr1c 1* | GATGAGA<u>CAT</u>GACATCTGTCACTTA | ATG start site | |
| *acvr1c 2* | TACTATTTGTCCTGTCTTACCTGG | e2i2 splice site | |
| *acvr1c 3* | TTAATGGGCACAGCCAGCTCTCACC | e3i3 splice site | |
| *acvr2a-a 1* | GCAGGTCC<u>CAT</u>TTTTTCACTCTTCT | ATG start site | *Albertson et al., 2005* |
| *acvr2a-a 2* | AGCAGTAGGGAATACCTGTCATAGC | e2i2 splice site | |

*Continued on next page*

*Continued*

| Target | Morpholino sequence (5'->3') | Target site | Reference |
|---|---|---|---|
| *acvr2a-a 3* | TCGCTGAATGGAGCCTTACTCTGAA | e3i3 splice site | |
| *acvr2a-b 1* | TCGATGGTCCCCGAGCGGTTCTTC | internal | |
| *acvr2a-b 2* | TGGCTGCACACAAACACAGATTAAT | splice site | **Dogra et al., 2017** |
| *acvr2a-b 3* | TGACAGAAGTATTTACCTGTGACGG | e3i3 splice site | |
| *acvr2b-a 1* | GCAGAGAAGCGAACATATTCCTTT | ATG start site | **Albertson et al., 2005** |
| *acvr2b-a 2* | TGAGCAGAGAAGCGAACATATTCCT | ATG start site | **Dogra et al., 2017** |
| *acvr2b-a 3* | AATGTTTAAGAGAGTCACCTGGTTC | e3i3 splice site | |
| *acvr2b-b* | AGCCAGCCAGGGAACAAACATATTC | ATG start site | **Dogra et al., 2017** |
| *control* | CCTCTTACCTCAGTTACAATTTATA | n.a. | Gene Tools |

The *acvr1b-b* transcriptional start site-targeting morpholino (MO-1) is complementary to the start codon and the region upstream of the *acvr1b-b* coding sequence. Therefore, *acvr1b-b* MO-1 does not target the *acvr1b-b* mRNA synthesized for rescue experiments (**Figure 4** and **Figure 4—figure supplement 1**).

## CRISPR F0 KO

F0 knockout (KO) embryos were generated using the CRISPR-Cas9 method described by **Kroll et al., 2021** following the protocol available at https://doi.org/10.17504/protocols.io.bs2rngd6. For each target gene, two or three synthetic guide RNAs (gRNAs), resulting from annealing a gene-specific crRNA to the Alt-R CRISPR-Cas9 tracrRNA (IDT, 1072532), were used. Each gRNA was assembled with Cas9 protein (Alt-R S.p. Cas9 Nuclease V3, IDT, 1081058) to a Cas9/gRNA ribonucleoprotein (RNP) and tested for its cutting efficiency using a T7 endonuclease I assay (see below, **Figure 2—figure supplement 3**; **Figure 3—figure supplement 3**). RNPs with low efficiency or toxic side effects were excluded from further experiments. RNPs targeting the same gene were subsequently pooled in equal amounts. For multi-gene KOs, respective RNP pools were mixed. Double *acvr1b-a,acvr1b-b* F0 KOs and quadruple *acvr2a-a,acvr2a-b,acvr2b-a,avcr2b-b* F0 KOs were generated by the injection of 2 nl RNP mix. In all other cases, an injection volume of 1 nl was used.

The following crRNAs, obtained from IDT, were used for F0 KO experiments:

| Target | Resulting RNP | Sequence | PAM |
|---|---|---|---|
| *acvr1b-a* | RNP2 | GAACCAGGAACGTTCCTCCC | TGG |
| | RNP3 | ACTACTCCGTCACAATCGAG | GGG |
| *acvr1b-b* | RNP1 | CAACTGGTGGCAGAGCTACG | AGG |
| | RNP2 | TGGAGGAGAGCATTAACATG | AGG |
| | RNP3 | GAGGAACCAGCTTCTCCTTG | TGG |
| *acvr2a-a* | RNP1 | CTCGTTCCACGTCACTACGT | TGG |
| | RNP3 | GCAACCTAGACATTGAGCTG | TGG |
| *acvr2a-b* | RNP1 | CGAGGACATTCCCAACCTGA | AGG |
| | RNP2 | TCTGAGAATCGACATGTACG | CGG |
| | RNP3 | GCTGTTCACTGACCTGGACA | CGG |
| *acvr2b-a* | RNP1 | GAAGACGAACCGTAGCGGTG | TGG |
| | RNP2 | GCGGGAGATGTTTTCCACTC | CGG |

*Continued on next page*

*Continued*

| Target | Resulting RNP | Sequence | PAM |
|--------|---------------|----------|-----|
| | RNP1 | ACGAGGCACCAACCTTCAGA | TGG |
| *acvr2b-b* | RNP2 | ATTTCGGCAAGCATGTCCCG | AGG |
| | RNP3 | AGAGGCTTCAGTCCAACCAG | AGG |
| | RNP1 | GCCTGTCCGAAGTACTTCAC | CGG |
| *oep* | RNP2 | TTCTGAACCCATTCTCCATG | TGG |
| | RNP3 | AAGAATTCAGCGTATTGCTT | TGG |

The *acvr1b-b* RNP1 has *acvr1b-a* as a predicted off-target with one mismatch in the gRNA's seed-sequence. Since the ultimate goal was a double *acvr1b-a,acvr1b-b* F0 KO, this potential off-target was accepted.

## T7E1 assay

To estimate the cutting efficiency of RNPs used for the F0 KOs, T7 endonuclease I (T7E1) assays were performed. To this end, genomic DNA was isolated from a pool of ten F0 KO embryos at 1 dpf using the 'hotshot' method (*Meeker et al., 2007*). A region of 190–290 bp surrounding the intended cut side was amplified using standard PCR conditions and the following primers:

| Target | RNP | Forward primer (5'->3') | Reverse primer (5'->3') |
|--------|-----|-------------------------|-------------------------|
| *acvr1b-a* | RNP2 | GTCTGAAAAGTGTTTTGCCTGTG | CAATGAAGCCCAAGATGTTTTCATG |
| | RNP3 | TTTCTTAGACAATGGCACATGGAC | CCTGGTTTCCCTGTAGGAGATC |
| | RNP1 | ATTCATGAAGAATATCAGCTGCCC | GCTCTTCTATGAAGCTGACGGT |
| *acvr1b-b* | RNP2 | CACTGTCATCATGGCAAAACAAC | CAAAGATTTGTTTTCAGCACCTCC |
| | RNP3 | GTTGTGTTTGCAGCTCTGTTGT | CTGTTGCAGTAGTCGGTGTAGC |
| *acvr2a-a* | RNP1 | GTTAATTGTATTGCAGGCTGATGC | CCTAAAAACGCAGACGAGACAG |
| | RNP3 | CTTAGGACAAACTGTCTTGGCAG | TAACGATATGTGATGGCAGAGGG |
| | RNP1 | TGTGTGTCTGTGTTCAGGGTTC | GGATGATGATGACGTTTCAGGTG |
| *acvr2a-b* | RNP2 | TTAATGCAATGATGTGTCTTTTGTGTG | TCTCTCTCTCTTACCGTCAGCC |
| | RNP3 | GTTCTGTGTTCAGGATCCAGGT | AGAAACCCAGAAATGTCAAAAGGC |
| *acvr2b-a* | RNP1 | GTGTGTTTGTTTACAGACCCCAG | CTGTCGTAGCAGTTGAAGTCGT |
| | RNP2 | CACTTTTGTCCAAATCGTCTGGT | CCAGAACTCCATCTCCAGGTTAG |
| | RNP1 | CATGGCAGAACGAAAGAGACATT | AAAATGCAGCCATTACGAGTTTTC |
| *acvr2b-b* | RNP2 | GTTCGCTGACAGATTACCTGAAG | ACTTTCTTTGGACGACCTACCAG |
| | RNP3 | AGAGGCATCCATATTTTCAAAGCAG | ACAGCCACATATTCGCTCAGTA |
| | RNP1 | GCGTTTGCAACCTTGTGTAATATC | TGGAATAACACCACAATCCCTGT |
| *oep* | RNP2 | CAGGAGCTGTGAATACGATGAAC | GAAGCAATGCAAAAGTCCATATCC |
| | RNP3 | AGCTGTTTCACTCGAGTCAGG | TTGCGTTAATGACAATCTCACCTC |

Without further purification, the resulting PCR product was annealed and digested with T7E1 (NEB, M0302) according to the manufacturer's instructions. For each sample, a 'no enzyme' (T7E1 -) control was included. The digested fragments were subsequently analyzed on 2% agarose gels.

## qRT-PCR

For qRT-PCR experiments, single embryos (*Figure 1E*) or groups of 10 embryos (*Figure 2—figure supplement 1E-F*; *Figure 3—figure supplement 1D*) were collected at shield stage, and total RNA was isolated using NucleoZol (Macherey-Nagel 740404.200) according to the manufacturer's protocol.

100 ng of RNA were used for cDNA synthesis with SuperScript III Reverse Transcriptase (Invitrogen 18080044) according to the manufacturer's protocol. qRT-PCR was performed with Platinum SYBR Green qPCR SuperMix-UDG (Invitrogen 1173304) on a CFX Connect Real-Time System (Bio-Rad 1855201). Two μl of 1:5 diluted cDNA were used as a template. The following primers were used for qRT-PCR analysis:

| Target | Forward primer (5'->3') | Reverse primer (5'->3') |
|---|---|---|
| eF1α | AGAAGGAAGCCGCTGAGATGG | TCCGTTCTTGGAGATACCAGCC |
| acvr1b-a | CGCCATGAAAACATCTTGG | GTGTCCATGTGCCATTGTCT |
| acvr1b-b | CTCTCCACCTCAGGATCAGG | GTACGAGCCACGGTCCTTT |
| acvr1c | GAGATTATTGGCACCCAAGG | AACCAGGATGTTCTTTGACTTTATG |
| acvr2a-a | GGTGTCCTCACAACATTG | TCACCGGTCACTCGACAC |
| acvr2a-b | GTGACACACACGGACAGGTT | AAACTGATCGCTCCTTCCAG |
| acvr2b-a | CAAACCAGCCATCGCACA | TCACACCAGTCTACGACC |
| acvr2b-b | ACACGTCGACATCGGACAG | AGGCTTCAGTCCAACCAGAG |

Transcript levels were normalized to the expression of the internal control eF1α using the $\Delta\Delta C_t$ method. Technical duplicates and biological triplicates were performed for each sample.

## Testing the splice-blocking *acvr1b-b* morpholino

To test the morpholino *acvr1b-b* 2 (MO-2) targeting the e7i7 splice site, embryos were injected with 0.4 ng or 2 ng MO-2 or left uninjected. In each case, 10 embryos at the 512 cell stage and 10 embryos at shield stage were used to prepare cDNA as described above. The cDNA subsequently served as a template to PCR-amplify fragments specific to (i) the unspliced *acvr1b-b* transcript still containing intron 7 and (ii) the spliced transcript. The PCR was conducted using KOD Hot Start DNA Polymerase (Novagen, 71086) according to the manufacturer's instructions with the following primers:

| Primer | Sequence | Binding site | Product size (bp) Spliced transcript | Unspliced transcript |
|---|---|---|---|---|
| (i) forward | CAAAAATGCCTACTGAGACAGCC | intron 7 | / | 334 |
| (ii) forward | ATGTGCTGATATCTACGCTCTGG | exon 7 | 171 | (2218) |
| reverse | ATGTTGGGTCGTAATCTCTGGTC | exon 8 | | |

An extension time of 10 s was used to suppress the 2218 bp PCR product expected for the PCR (ii) in the case of unspliced *acvr1b-b* transcripts.

## pSmad2/3 and pSmad2/3-pSmad1/5/9 double immunostainings

Embryos were fixed in 4% formaldehyde in PBS overnight at 4 °C, dehydrated in 100% methanol and stored at –20 °C. For pSmad2/3 immunofluorescence stainings, fixed embryos were incubated in acetone for 7 min, washed three times for 5 min with PBST (PBS +0.1% Tween 20), blocked for at least 1 hr with 10% FBS (Biochrom S0415) in PBST and incubated with 1:5000 rabbit anti-phospho-Smad2/Smad3 primary antibody (Cell Signaling Technologies 8828, RRID: AB_2631089) in blocking solution at 4 °C overnight. The following day, embryos were washed 8 times for 15 min with PBST, blocked for at least 1 hr with blocking solution and incubated with 1:500 goat anti-rabbit horseradish peroxidase secondary antibody (Jackson ImmunoResearch 111-035-003, RRID: AB_2313567) in blocking solution at 4 °C overnight. Embryos were then washed eight times for 15 min with PBST, incubated in TSA 1×amplification buffer (TSA Plus Fluorescein Kit, Perkin Elmer, NEL741001KT) for 15 min and stained by incubation in 75 μl 1:75 fluorescein-TSA in 1×amplification buffer for 45 min. Embryos were washed three times for 5 min with PBST, 30 min with methanol and washed twice more with PBST before incubating them in 1:5000 DAPI in PBST at room temperature (RT) for at least 1 hr, followed by at least three washes with PBST. Embryos were then transferred into methanol and stored at –20 °C before imaging.

Dual staining of pSmad2/3 with pSmad1/5/9 was performed as described previously (*Soh et al., 2020*). Staining of pSmad2/3 with fluorescein-TSA and subsequent washes with PBST (see above) were followed by a 3 hr incubation in methanol. The embryos were then washed three times for 10 min in PBST and blocked for at least 1 hr with blocking solution before being incubated with 1:100 rabbit anti-phospho-Smad1/Smad5/Smad9 primary antibody (Cell Signaling Technologies 13820 S, RRID: AB_2493181) in blocking solution at 4 °C overnight. Following ten PBST washes of 15 min each and at least 1 hr in blocking solution, the embryos were incubated with 1:100 anti-rabbit Alexa647 IgG (Invitrogen A21245, RRID:AB_141775) in blocking solution at 4 °C overnight. The next day, embryos were washed eight times with PBST for 15 min each before being imaged using a light-sheet microscope.

## Imaging

Brightfield images for the documentation of embryo morphology were taken using an Axio Zoom. V16 (ZEISS) microscope with a PlanNeoFluar Z 1×objective or a M205 FCA (Leica) microscope with a Planapo 1.0×M-Series objective.

Fluorescence images of fixed and live embryos were obtained using a Lightsheet Z.1 microscope (ZEISS). For mounting, the samples were drawn into 1.5% low-melting point agarose (Lonza 50080) with a size 3 glass capillary sample holder (ZEISS). If not noted otherwise, embryos were imaged using a W Plan-Apochromat 20×objective with 0.7×zoom and 5 µm intervals between z-slices. For imaging of pSmad2/3 immunostainings (*Figure 4* and *Figure 6*), embryos were imaged from different angles using a 488 nm laser at 2% power with 100 ms exposure time. For DAPI stainings, embryos were imaged using a 405 nm laser at 10% laser power with 70 ms exposure time. For YSL injections, z-stacks comprising 15 slices were taken using a 488 nm laser with 100% laser power and 70 ms exposure time to image GFP and a 638 nm far-red laser at 1% laser power and 20 ms exposure time to detect Alexa Fluor 647 dextran. pSmad2/3-pSmad1/5/9 double immunostainings (*Figure 2* and *Figure 3*) and transplanted Nodal clones (*Figure 5*) were imaged using a W Plan-Apochromat 10×objective with 1×zoom and 1.82 µm intervals between z-slices. For pSmad2/3-pSmad1/5/9 double immunostainings, a laser power of 1% and 10% was used for the 488 nm and the 638 nm laser, respectively, with an exposure time of 70 ms. For transplanted Nodal clones, 488 nm and 638 nm lasers were used at a power of 15% and 10%, respectively, and 250 ms exposure time.

FRAP and FDAP measurements were performed using an LSM 780 NLO confocal microscope (ZEISS) with an LD C-Apochromat 40×/1.1 W Korr objective. Embryos were mounted in 1.5% low-melting point agarose in glass-bottom petri dishes (MatTek Corporation P35G-1.5–20 C). After solidification, the agarose was covered with embryo medium to protect the embryos from drying out. FRAP and FDAP measurements were performed and analyzed as previously described (*Bläßle and Müller, 2015*; *Bläßle et al., 2018*; *Müller et al., 2012*; *Rogers et al., 2015*; *Soh and Müller, 2018*). FRAP and FDAP data sets that were poorly fit by the diffusion-production-clearance model (overall $R^2$ <0.8, high local variability, linear increase, or severe mismatch between early recovery kinetics) or the exponentially decreasing function (overall $R^2$ <0.88 or bleed-through artifacts), respectively, were excluded. Within the FDAP data sets, single frames were excluded in rare cases of signal saturation or signal artifacts, for example from transient bubbles in the oil resulting from multi-position imaging.

## Quantification of pSmad2/3 immunofluorescence signal levels

Levels of pSmad2/3 immunofluorescence were quantified on maximum intensity projections of samples recorded on the same day using Fiji (*Schindelin et al., 2012*). A mask generated from the DAPI channel was used to exclude non-nuclear pSmad2/3 signals. Manually drawn regions of interest on the dorsal or lateral side were used to obtain line intensity profiles from the embryo's margin to the animal pole in the corresponding domain. Background subtraction was performed for each embryo individually using the median intensity of a region close to the animal pole. The profiles were normalized to the value of the average wild-type marginal zone signal on the lateral side.

## YSL-injection image analysis

Images obtained from embryos that had been YSL-injected with Nodal-encoding mRNA were analyzed using Fiji (*Schindelin et al., 2012*). To exclude fluorescent signal in the YSL, the far-red channel was converted into a mask with the mean thresholding algorithm in Fiji. Ten marginal z-slices of the GFP channel were then used for a maximum intensity projection. Before the region of interest around

the embryo was defined, the maximum intensity projections were rotated, so that the YSL was on the left, parallel to the image margin. Pixels outside of the embryo and bright staining artifacts were set to *n.a.* to avoid distortion of the calculated averages. For Squint-GFP, the *plot profile* function in Fiji was used to extract the averaged intensities from the embryo. Background levels determined by measuring uninjected embryos were subtracted from the gradient profiles. The profiles were normalized following previously described procedures (*Gregor et al., 2007*; *Rogers et al., 2020*) with the model $I_n\left(x\right) = A_n \bar{c}\left(x\right) + b_n$ , relating the mean intensity profile $\bar{c}\left(x\right)$ of all data points to each embryo's intensity profile $I_n\left(x\right)$ through the embryo-specific proportionality constant $A_n$ and the non-specific background $b_n$ . $A_n$ and $b_n$ were determined by minimizing the sum of squared differences between the model and the intensity profiles using the function *fminsearch* in MATLAB 7.10.0 (*Rogers et al., 2020*). Finally, each profile was normalized to the intensity at a distance of 0 µm from the YSL.

The Fiji *find maxima* function was used to identify Cyclops-GFP puncta. Uninjected embryos were used to verify that this approach only identified single maxima in order to exclude artifacts. The x- and y-coordinates of the puncta were extracted using the function *measure*, and the distribution of puncta as a function of distance from the YSL was plotted.

## Transplantation of Cyclops-GFP or Squint-GFP clones

Ectopic Nodal signaling sources were generated by transplanting Cyclops-GFP or Squint-GFP expressing clones from donor embryos into wild-type, MZ*oep*, or Type I receptor loss-of-function (*acvr1b-a*-/-+0.4 ng *acvr1b-b* MO-1) host embryos. Donor embryos were generated by the injection of *squint-GFP* or *cyclops-GFP* mRNA and Alexa Fluor 647 dextran into one-cell stage wild-type embryos (see *Microinjections and embryo dechorionation*). For control experiments, donor embryos injected only with Alexa Fluor 647 dextran were used. Dechorionated sphere-stage host and donor embryos were transferred to Ringer's solution (116 mM NaCl, 2.9 mM KCl, 1.8 mM CaCl$_2$, 5 mM HEPES pH 7.2) and animal pole to animal pole cell transplantation was performed as described previously (*Soh et al., 2021*; *Soh et al., 2020*). Typically, two clones were derived from each donor embryo. After transplantation, embryos were kept in Ringer's solution for 60 min (30 min at RT and 30 min at 28 °C).

To determine the distribution of Squint-GFP and Cyclops-GFP, host embryos were mounted in 1.5% low-melting point agarose (Lonza) with the animal-vegetal axis orthogonal to the agarose column and imaged using a Lightsheet Z.1 microscope (ZEISS) 60 min post-transplantation (see *Imaging*). The dispersal of Squint-GFP and Cyclops-GFP secreted from the clone into the host embryos was measured using Fiji (*Schindelin et al., 2012*). To this end, maximum intensity projections of 15 z-slices (~27 µm) were generated covering the embryo's animal pole at a depth of approximately 51 µm – 78 µm. Linear regions-of-interest of ~70 µm width and variable length were drawn radially from the edge of the transplanted clone (determined as an intensity value of 700 in the far-red channel) towards the embryo's outline. GFP signal intensity profiles were obtained from these regions-of-interest and averaged for each embryo. Background levels were determined as the median of GFP intensities obtained from control transplantations performed on the same day and subtracted from the intensity profiles. The background-subtracted intensity profile of each embryo was then normalized to the intensity at a distance of 0 µm from the transplantation site. Embryos with leaking yolk or a damaged clone were excluded from the analysis (3 out of 45). For image presentation (*Figure 5B*), Sébastien Tosi's Fiji macro "deStripe2" (*Tosi, 2020*) was used.

## Transplantation of marginal cells

Donor embryos were obtained from an H2A.F/Z:GFP incross. Wild-type TE as well as MZ*sqt*-/-;*cyc*-/- host embryos were collected 1 hr later. Only H2A.F/Z:GFP embryos exhibiting strong fluorescence were used as donors. The embryos were transferred to Ringer's solution (116 mM NaCl, 2.9 mM KCl, 1.8 mM CaCl$_2$, 5 mM HEPES pH 7.2) for margin transplantations. Margin cells were taken from donors around the 30–40% epiboly stage and transplanted into the animal pole or the marginal region of hosts (hosts were around sphere stage) as described in *Soh et al., 2021* using glass needles with an inner tip diameter of ~80–90 µm. Typically, two margin transplants were derived from each donor embryo (taken from opposing regions). To keep the experimental groups as similar as possible, transplantations were performed such that TE and MZ*sqt*-/-;*cyc*-/- embryos were used as hosts in an alternating manner. The embryos were kept in Ringer's solution for 30 min at RT, transferred to embryo medium at 28 °C and then fixed 2 hr post-transplantation in PBS with 4% formaldehyde.

After overnight fixation at 4 °C, embryos were processed for pSmad2/3 immunostainings as described above and additionally used for GFP immunostainings with 1:1000 anti-GFP antibody (Aves Labs #GFP-1020, RRID: AB_10000240) at 4 °C overnight. The samples were briefly rinsed with PBST and then washed six times for 20 min each before blocking with 500 µl blocking solution for 1.5 hr. The blocking solution was removed, and a 1:500 dilution of Alexa Fluor 568-conjugated anti-chicken IgY (Abcam 175477) in blocking solution was added to the samples, which were then kept shaking at 4 °C overnight. PBST was added to briefly rinse the samples, and the samples were then washed twelve times for approximately 20 min each. They were stored in PBST containing 1 mg/l DAPI at 4 °C until imaging on a Lightsheet Z.1 microscope (ZEISS) with a W Plan-Apochromat 20×/1.0 objective. The samples were mounted in 1.5% low-melting point agarose (Lonza) in embryo medium and imaged in water. All samples and controls from one experiment were imaged on the same day to ensure comparable fluorescence between embryos. The embryos were mounted with the animal-vegetal axis orthogonal (margin-to-animal pole transplantations) or parallel (margin-to-margin transplantations) to the agarose column. z-stacks covering 130 µm from the animal pole were acquired (13 slices with 10 µm steps), and maximum intensity projections over 110 µm (ignoring the two animal-most slices) were generated using Fiji (*Schindelin et al., 2012*). The outline of the transplants was drawn around cells that exhibited immunofluorescence signal for GFP. For each experimental setup, three independent transplantation experiments were performed on 2 days. All fixed samples per experimental setup were immunostained in parallel.

To measure the distribution of the pSmad2/3 signal extending from the transplants into the host embryos, signal intensity profiles of manually drawn rectangular regions of interest were obtained using Fiji (*Schindelin et al., 2012*). These regions of interest were 100 µm in height and variable in width, ranging from the edge of the transplanted cell cluster to the animal pole in the case of margin transplants or the direction of maximal pSmad2/3 signal distribution for animal pole transplants. Background levels, determined from embryonic regions without nuclear pSmad2/3 signal, were subtracted from the profiles. To account for differences in transplant size, individual profiles were normalized to the number of transplanted cells. The number of transplanted cells was determined as the number of GFP-positive nuclei within the z-stack using the particle detection of TrackMate (*Tinevez et al., 2017*), an open-source plugin for Fiji (*Schindelin et al., 2012*). To this end, spots of an estimated diameter of 8 µm were detected and subsequently filtered based on a manually adapted quality threshold. Remaining spurious spots (for example, located outside of the embryo) were removed manually.

The pSmad2/3 intensities within the transplants were measured in a circular region of defined size using Fiji (*Schindelin et al., 2012*). The intensities in *Figure 6—figure supplement 2D* are given relative to the mean wild-type intensity. To count animal pole-facing pSmad2/3 positive nuclei (*Figure 6—figure supplement 2E*), a line parallel to the margin was drawn just above the animal-most transplanted nucleus. pSmad2/3 positive nuclei on the animal side of this line were counted.

## Statistical analysis

p-values for differences between experimental conditions were calculated using two-tailed Student's t-tests assuming equal variance in Excel for *Figure 1*, *Figure 2—figure supplement 1*, *Figure 3—figure supplement 1*, *Figure 4*, *Figure 6*, and *Figure 6—figure supplement 2*. Since an F test in R (*R Development Core Team, 2017*) showed that the two experimental conditions in *Figure 6—figure supplement 2D* and *Figure 3—figure supplement 1* did not have equal variance, a Student's t-test with unequal variance was performed in Excel for this data. A Shapiro-Wilk test in R showed that the data in *Figure 6—figure supplement 2E* was not normally distributed, and a Wilcoxon rank sum test was therefore performed to calculate a p-value (note that due to the presence of ties, the p-value is not exact in this case but a normal approximation). Cohen's *d* (Hedges bias corrected) as a measure for effect size was determined using Robert Coe's Effect Size Calculator (*Coe, 2002*).

## p-values for Figure 1E

| | | *oep* | | |
|---|---|---|---|---|
| | +*squint-GFP* | +*cyclops-GFP* | +*lefty-D2* | +SB-505124 |
| Uninjected | 0.005 | <0.001 | 0.003 | 0.631 |

| | | *acvr1b-a* | | |
|---|---|---|---|---|
| | +*squint-GFP* | +*cyclops-GFP* | +*lefty-D2* | +SB-505124 |
| Uninjected | <0.001 | <0.001 | 0.003 | 0.001 |

| | | *acvr1b-b* | | |
|---|---|---|---|---|
| | +*squint-GFP* | +*cyclops-GFP* | +*lefty-D2* | +SB-505124 |
| Uninjected | 0.503 | 0.587 | 0.395 | 0.980 |

| | | *acvr2a-a* | | |
|---|---|---|---|---|
| | +*squint-GFP* | +*cyclops-GFP* | +*lefty-D2* | +SB-505124 |
| Uninjected | 0.188 | 0.401 | 0.419 | 0.705 |

| | | *acvr2a-b* | | |
|---|---|---|---|---|
| | +*squint-GFP* | +*cyclops-GFP* | +*lefty-D2* | +SB-505124 |
| Uninjected | 0.014 | 0.278 | 0.777 | 0.883 |

| | | *acvr2b-a* | | |
|---|---|---|---|---|
| | +*squint-GFP* | +*cyclops-GFP* | +*lefty-D2* | +SB-505124 |
| Uninjected | 0.108 | 0.110 | 0.182 | 0.920 |

| | | *acvr2b-b* | | |
|---|---|---|---|---|
| | +*squint-GFP* | +*cyclops-GFP* | +*lefty-D2* | +SB-505124 |
| Uninjected | 0.101 | 0.260 | 0.897 | 0.797 |

## p-values for Figure 2—figure supplement 1E, F

| | *acvr2a-b* | |
|---|---|---|
| | $acvr2a\text{-}a^{SA34654}$ | $acvr2b\text{-}a^{t08pm}$ |
| Wild type | 0.076 | 0.016 |

| | *acvr2b-b* | |
|---|---|---|
| | $acvr2a\text{-}a^{SA34654}$ | $acvr2b\text{-}a^{t08pm}$ |
| Wild type | 0.068 | 0.081 |

## p-values for Figure 3—figure supplement 1D

| | *acvr1b-b* |
|---|---|
| | $acvr1b\text{-}a^{t03pm}$ |
| Wild type | 0.376 |

## p-values for Figure 4B

| | $acvr1b\text{-}a^{-/-}$ | Wild type $+acvr1b\text{-}b$ MO-1 | $acvr1b\text{-}a^{-/-}$ $+acvr1b\text{-}b$ MO-1 | Wild type $+acvr1b\text{-}b$ MO-1 $+acvr1b\text{-}a$ mRNA | $acvr1b\text{-}a^{-/-}$ $+acvr1b\text{-}b$ MO-1 $+acvr1b\text{-}a$ mRNA | Wild type $+acvr1b\text{-}b$ MO-1 $+acvr1b\text{-}b$ mRNA | $acvr1b\text{-}a^{-/-}$ $+acvr1b\text{-}b$ MO-1 $+acvr1b\text{-}b$ mRNA |
|---|---|---|---|---|---|---|---|
| Wild type | 0.203 | 0.785 | <0.001 | 0.783 | 0.041 | 0.286 | 0.097 |

## p-values and Cohen's *d* for Figure 6A

| | Squint-Dendra2 | | Cyclops-Dendra2 | |
|---|---|---|---|---|
| | $acvr1b\text{-}a^{-/-}$ $+acvr1b\text{-}b$ MO-1 | Wild type $+acvr2b\text{-}a$ mRNA | $acvr1b\text{-}a^{-/-}$ $+acvr1b\text{-}b$ MO-1 | Wild type $+acvr2b\text{-}a$ mRNA |
| Wild type | 0.458, 0.30 | 0.147, 0.73 | 0.006, 1.99 | 0.175, 0.82 |

## p-values and Cohen's *d* for Figure 6B

| | Squint-GFP | | |
|---|---|---|---|
| | Wild type $+oep$ mRNA | $acvr1b\text{-}a^{-/-}$ $+acvr1b\text{-}b$ MO-1 | Wild type $+acvr2b\text{-}a$ mRNA |
| Wild type | 0.003, −1.40 | 0.028, 1.03 | 0.531, −0.31 |

## p-value and Cohen's *d* for Figure 6—figure supplement 2C

| | $MZsqt^{-/-};cyc^{-/-}$ | |
|---|---|---|
| | Animal pole transplants | Margin transplants |
| Wild type | 0.201, 0.42 | 0.697, −0.13 |

## p-value and Cohen's *d* for Figure 6—figure supplement 2D

| | $MZsqt^{-/-};cyc^{-/-}$ |
|---|---|
| Wild type | <0.001, 2.21 |

## p-value and Cohen's *d* for Figure 6—figure supplement 2E

| | $MZsqt^{-/-};cyc^{-/-}$ |
|---|---|
| Wild type | 0.148, 0.56 |

## Acknowledgements

We thank Christine Henzler, Sarah Keim, Jens Dominik Maile and Hannah Wild for technical support and discussions. We are grateful to the European Zebrafish Research Center (EZRC) for providing the *acvr2a-a*[sa34654] and *acvr2a-b*[sa18285] zebrafish lines. This project has received funding from the European Research Council (ERC) under the European Union's Horizon 2020 research and innovation program (grant agreement No 637840 (QUANTPATTERN) and grant agreement No 863952 (ACE-OF-SPACE)). This work was also funded by the Max Planck Society and the International Max Planck Research School "From Molecules to Organisms".

## Additional information

### Funding

| Funder | Grant reference number | Author |
| --- | --- | --- |
| International Max Planck Research School "From Molecules to Organisms" | Graduate Student Fellowship | Hannes Preiß David Mörsdorf |
| Max Planck Society | Max Planck Research Group | Patrick Müller |
| European Research Council | Grant agreement No 637840 (QUANTPATTERN) | Patrick Müller |
| European Research Council | Grant agreement No 863952 (ACE-OF-SPACE) | Patrick Müller |

The funders had no role in study design, data collection and interpretation, or the decision to submit the work for publication.

### Author contributions

Hannes Preiß, Conceptualization, Resources, Data curation, Formal analysis, Investigation, Visualization, Methodology, Writing – original draft, Writing – review and editing; Anna C Kögler, Conceptualization, Resources, Data curation, Formal analysis, Validation, Investigation, Visualization, Methodology, Writing – original draft, Project administration, Writing – review and editing; David Mörsdorf, Conceptualization, Data curation, Formal analysis, Investigation, Visualization, Methodology, Writing – original draft, Writing – review and editing; Daniel Čapek, Katherine W Rogers, Data curation, Formal analysis, Validation, Investigation, Visualization, Writing – review and editing; Gary H Soh, Resources, Data curation, Validation, Investigation, Methodology, Writing – review and editing; Hernán Morales-Navarrete, Software, Formal analysis, Validation, Visualization, Writing – review and editing; María Almuedo-Castillo, Resources, Data curation, Investigation, Writing – review and editing; Patrick Müller, Conceptualization, Resources, Software, Formal analysis, Supervision, Funding acquisition, Validation, Investigation, Visualization, Methodology, Writing – original draft, Project administration, Writing – review and editing

### Author ORCIDs

Hannes Preiß http://orcid.org/0000-0001-6873-9440
Anna C Kögler http://orcid.org/0000-0003-2794-3589
David Mörsdorf http://orcid.org/0000-0001-8982-2155
Daniel Čapek http://orcid.org/0000-0001-5199-9940
Gary H Soh http://orcid.org/0000-0003-0755-6805
Katherine W Rogers http://orcid.org/0000-0001-5700-2662
Hernán Morales-Navarrete http://orcid.org/0000-0002-9578-2556
María Almuedo-Castillo http://orcid.org/0000-0001-6759-5879
Patrick Müller http://orcid.org/0000-0002-0702-6209

### Ethics

All procedures were executed in accordance with the guidelines of the State of Baden-Württemberg and approved by the Regierungspräsidium Tübingen and the Regierungspräsidium Freiburg.

### Decision letter and Author response

Decision letter https://doi.org/10.7554/eLife.66397.sa1
Author response https://doi.org/10.7554/eLife.66397.sa2

## Additional files

### Supplementary files

• Transparent reporting form

## Data availability

Source data files containing the numerical data used to generate the figures have been provided.

The following previously published dataset was used:

| Author(s) | Year | Dataset title | Dataset URL | Database and Identifier |
|---|---|---|---|---|
| Collins JE, Wali N, Dooley CM, Busch-Nentwich EM | 2016 | Baseline_expression_from_transcriptional_profiling_of_zebrafish_developmental_stages | http://www.ebi.ac.uk/ena/data/view/PRJEB7244 | EBI, PRJEB7244 |

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

# Appendix 1

## Appendix 1—key resources table

| Reagent type (species) or resource | Designation | Source or reference | Identifiers | Additional information |
|---|---|---|---|---|
| Antibody | Anti-chicken IgY Alexa Fluor 568-conjungated (Goat polyclonal) | Abcam | 175477 | 1:500 |
| Antibody | Anti-Digoxigenin-AP, Fab-Fragmente (Sheep polyclonal) | Sigma-Aldrich | Roche-11093274910, RRID:AB_2734716 | 1:5000 |
| Antibody | Anti-GFP antibody (Chicken polyclonal) | Aves Lab | #GFP-1020, RRID:AB_10000240 | 1:1000 |
| Antibody | Anti-phospho-Smad2/Smad3 (Rabbit monoclonal) | Cell Signaling Technology | 8828, RRID:AB_2631089 | 1:5000 |
| Antibody | Anti-phospho-Smad1/Smad5/Smad9 (Rabbit monoclonal) | Cell Signaling Technologies | 13820 S, RRID:AB_2493181 | 1:100 |
| Antibody | Anti-rabbit Alexa647 IgG (Goat polyclonal) | Invitrogen | A21245, RRID:AB_141775 | 1:100 |
| Antibody | Anti-rabbit horseradish peroxidase (Goat polyclonal) | Jackson ImmunoResearch | 111-035-003, RRID:AB_2313567 | 1:500 |
| Chemical compound, drug | DAPI | Life Technologies | D1306 | 1:5000 |
| Chemical compound, drug | FBS | Biochrom | S0415 | |
| Chemical compound, drug | Nodal inhibitor SB-505124 | Sigma | S4696-5MG | 10 µM |
| Chemical compound, drug | NucleoZol | Macherey-Nagel | 740404.200 | |
| Chemical compound, drug | Pronase | Roche | 11459643001 | |
| Commercial assay, kit | 5'/3' RACE Kit, 2$^{nd}$ Generation | Roche | 03353621001 | |
| Commercial assay, kit | pCR-bluntII TOPO kit | Thermo Fisher Scientific | 450245 | |
| Commercial assay, kit | Platinum SYBR Green qPCR SuperMix-UDG | Invitrogen | 1173304 | |
| Commercial assay, kit | RNeasy MinElute Cleanup kit | Qiagen | 74204 | |
| Commercial assay, kit | RNeasy Mini kit | Qiagen | 74104 | |
| Commercial assay, kit | SP6 mMessage | Thermo Fisher Scientific | AM1340 | |
| Commercial assay, kit | SuperScript III Reverse Transcriptase | Invitrogen | 18080044 | |
| Commercial assay, kit | TSA Plus Fluorescein Kit | Perkin Elmer | NEL741001KT | TSA plus cyanine 3 |
| Commercial assay, kit | Wizard SV Gel and PCR Clean-Up System | Promega | A9282 | |
| Gene (*Danio rerio*) | *acvr1b-a* | zfin.org | ZDB-GENE-980526–527 | |
| Gene (*Danio rerio*) | *acvr1b-b* | zfin.org | ZDB-GENE-131120–9 | |
| Gene (*Danio rerio*) | *acvr1c* | zfin.org | ZDB-GENE-181207–1 | |

*Appendix 1 Continued on next page*

*Appendix 1 Continued*

| Reagent type (species) or resource | Designation | Source or reference | Identifiers | Additional information |
|---|---|---|---|---|
| Gene (*Danio rerio*) | acvr2a-a | zfin.org | ZDB-GENE-980526–227 | |
| Gene (*Danio rerio*) | acvr2a-b | zfin.org | ZDB-GENE-120215–23 | |
| Gene (*Danio rerio*) | acvr2b-a | zfin.org | ZDB-GENE-980526–549 | |
| Gene (*Danio rerio*) | acvr2b-b | zfin.org | ZDB-GENE-170621–5 | |
| Other | Alexa Fluor 647 dextran | Invitrogen | D22914 | See Materials and methods, *Microinjections and embryo dechorionation* |
| Other | Low-melting point agarose | Lonza | 50080 | Mounting of samples for lightsheet microscopy |
| Other | Human ACVR1B | Uniprot | P36896 | Protein sequence |
| Other | Human ACVR1C | Uniprot | Q8NER5 | Protein sequence |
| Other | Human ACVR2A | Uniprot | P27037 | Protein sequence |
| Other | Human ACVR2B | Uniprot | Q13705 | Protein sequence |
| Other | Mouse ACVR1B | Uniprot | Q61271 | Protein sequence |
| Other | Mouse ACVR1C | Uniprot | Q8K348 | Protein sequence |
| Other | Mouse ACVR2A | Uniprot | P27038 | Protein sequence |
| Other | Mouse ACVR2B | Uniprot | P27040 | Protein sequence |
| Other | Zebrafish Acvr1b-a | Uniprot | P79689 | Protein sequence |
| Other | Zebrafish Acvr1b-b | Uniprot | F1QZF0 | Protein sequence |
| Other | Zebrafish Acvr1c | Uniprot | E7F0I3 | Protein sequence |
| Other | Zebrafish Acvr2a-a | Uniprot | F1QG01 | Protein sequence |
| Other | Zebrafish Acvr2a-b | Uniprot | A0A0R4ISZ9 | Protein sequence |
| Other | Zebrafish Acvr2b-a | Uniprot | Q9YGU4 | Protein sequence |
| Other | Zebrafish Acvr2b-b | Uniprot | A0A0G2KN81 | Protein sequence |
| Peptide, recombinant protein | BamHI | NEB | R3136 | |
| Peptide, recombinant protein | Alt-R S.p. Cas9 Nuclease V3 | IDT | 1081058 | |
| Peptide, recombinant protein | ClaI-HF | NEB | R0197 | |
| Peptide, recombinant protein | EcoRI-HF | NEB | R3101 | |
| Peptide, recombinant protein | Exonuclease I (ExoI) | NEB | M0568 | |
| Peptide, recombinant protein | NotI-HF | NEB | R3189 | |
| Peptide, recombinant protein | Recombinant Shrimp Alkaline Phospatase (rSAP) | NEB | M0371L | |

*Appendix 1 Continued on next page*

*Appendix 1 Continued*

| Reagent type (species) or resource | Designation | Source or reference | Identifiers | Additional information |
|---|---|---|---|---|
| Peptide, recombinant protein | StuI (Eco147I) | Thermo Fisher | FD0424 | |
| Peptide, recombinant protein | T7 endonuclease I (T7E1) | NEB | M0302 | |
| Peptide, recombinant protein | XbaI | NEB | R0145 | |
| Peptide, recombinant protein | XhoI | NEB | R0146 | |
| Recombinant DNA reagent | Cas9 nuclease (*Streptococcus pyogenes*) from CMV and T7 promoters | *Hwang et al., 2013* | MLM3613 RRID:Addgene_42251 | |
| Recombinant DNA reagent | pCS2+-*acvr1b-a* (plasmid) | Generated in this study | | See Materials and methods, *mRNA synthesis* |
| Recombinant DNA reagent | pCS2+-*acvr1b-b* (plasmid) | Generated in this study | | See Materials and methods, *mRNA synthesis* |
| Recombinant DNA reagent | pCS2+-*acvr1c* (plasmid) | Generated in this study | | See Materials and methods, *mRNA synthesis* |
| Recombinant DNA reagent | pCS2+-*acvr2a-a* (plasmid) | Generated in this study | | See Materials and methods, *mRNA synthesis* |
| Recombinant DNA reagent | pCS2+-*acvr2a-b* (plasmid) | Generated in this study | | See Materials and methods, *mRNA synthesis* |
| Recombinant DNA reagent | pCS2+-*acvr2b-a* (plasmid) | Generated in this study | | See Materials and methods, *mRNA synthesis* |
| Recombinant DNA reagent | pCS2+-*acvr2b-b* (plasmid) | Generated in this study | | See Materials and methods, *mRNA synthesis* |
| Recombinant DNA reagent | TOPO-*acvr1b-a* (plasmid) | Generated in this study | | See Materials and methods, *Whole-mount in situ hybridization* |
| Recombinant DNA reagent | TOPO-*acvr1b-b* (plasmid) | Generated in this study | | See Materials and methods, *Whole-mount in situ hybridization* |
| Recombinant DNA reagent | TOPO-*acvr1c* (plasmid) | Generated in this study | | See Materials and methods, *Whole-mount in situ hybridization* |
| Recombinant DNA reagent | TOPO-*acvr2a-a* (plasmid) | Generated in this study | | See Materials and methods, *Whole-mount in situ hybridization* |
| Recombinant DNA reagent | TOPO-*acvr2a-b* (plasmid) | Generated in this study | | See Materials and methods, *Whole-mount in situ hybridization* |
| Recombinant DNA reagent | TOPO-*acvr2b-a* (plasmid) | Generated in this study | | See Materials and methods, *Whole-mount in situ hybridization* |
| Recombinant DNA reagent | TOPO-*acvr2b-b* (plasmid) | Generated in this study | | See Materials and methods, *Whole-mount in situ hybridization* |
| Sequence-based reagent | Morpholino antisense oligo control | Gene Tools, LLC | | |
| Sequence-based reagent | Gene-specific crRNA | IDT | | |

*Appendix 1 Continued on next page*

*Appendix 1 Continued*

| Reagent type (species) or resource | Designation | Source or reference | Identifiers | Additional information |
|---|---|---|---|---|
| Sequence-based reagent | Alt-R CRISPR-Cas9 tracrRNA | IDT | 1072532 | |
| Software, algorithm | Blast | Uniprot | RRID:SCR_002380 | |
| Software, algorithm | CFX Manager Software | Bio-Rad | 1845000, RRID:SCR_017251 | |
| Software, algorithm | CHOPCHOP | *Montague et al., 2014* | RRID:SCR_015723 | |
| Software, algorithm | Clustal Omega | *Madeira et al., 2019* | RRID:SCR_001591 | |
| Software, algorithm | Excel | Microsoft | RRID:SCR_016137 | |
| Software, algorithm | Fiji | *Schindelin et al., 2012* | RRID:SCR_002285 | |
| Software, algorithm | Hetindel, Version 1 | https://github.com/najasplus/hetindel_shinyapp | RRID:SCR_018922 | |
| Software, algorithm | Jalview version 2.10.3b1 | *Waterhouse et al., 2009* | RRID:SCR_006459 | |
| Software, algorithm | Lasergene Seqman Pro 14 | DNASTAR | | |
| Software, algorithm | Matlab | Mathworks | http://mathworks.com RRID: SCR_001622 | |
| Software, algorithm | PolyPeakParser | *Hill et al., 2014* | | |
| Software, algorithm | PyFDAP | *Bläßle and Müller, 2015* | RRID:SCR_022664 | |
| Software, algorithm | PyFRAP | *Bläßle et al., 2018* | RRID:SCR_022665 | |
| Software, algorithm | R | R-Core-Team | RRID:SCR_001905 | |
| Strain, stain background (*E. coli*) | DH5α | In-house | | Chemically competent |
| Strain, strain background (*Danio rerio*) | acvr1b-a$^{t03pm}$ zebrafish mutant | Generated in this study | | See Materials and methods, *Fish lines and husbandry* |
| Strain, strain background (*Danio rerio*) | acvr1c$^{t06pm}$ zebrafish mutant | Generated in this study | | See Materials and methods, *Fish lines and husbandry* |
| Strain, strain background (*Danio rerio*) | acvr2a-a$^{sa34654}$ zebrafish mutant | EZRC | sa34654 | |
| Strain, strain background (*Danio rerio*) | acvr2a-b$^{sa18285}$ zebrafish mutant | EZRC | sa18285 | |
| Strain, strain background (*Danio rerio*) | acvr2b-a$^{t08pm}$ zebrafish mutant | Generated in this study | | See Materials and methods, *Fish lines and husbandry* |
| Strain, strain background (*Danio rerio*) | H2A.F/Z:GFP | *Pauls et al., 2001* | | |

*Appendix 1 Continued on next page*

*Appendix 1 Continued*

| Reagent type (species) or resource | Designation | Source or reference | Identifiers | Additional information |
|---|---|---|---|---|
| Strain, strain background (*Danio rerio*) | MZ*sqt*$^{-/-}$;*cyc*$^{-/-}$ | *Feldman et al., 1998*; *Schier et al., 1996*; *Ciruna et al., 2002* | | Mutants were obtained by germ line transplantation |
| Strain, strain background (*Danio rerio*) | *oep*$^{tz57}$ zebrafish mutant | *Gritsman et al., 1999*; *Zhang et al., 1998* | tz57 | |
| Strain, strain background (*Danio rerio*) | TE zebrafish | In-house | | Wild type |

