## [Editor Report]

While the Nodal signaling pathway plays essential roles in germ layer induction and patterning during development, the precise function of different zebrafish Nodal receptors had not been determined. Here, the authors identify seven genes encoding Type I and Type II Nodal receptors in zebrafish and use a combination of genetic approaches to show that the Type I receptors Acvr1b-a and Acvr1b-b act redundantly as major mediators of Nodal signaling. Whereas the Acvr2 Type II receptors function partly redundantly and partially independently of Nodal in embryo patterning. Moreover, Type I receptor and co-receptor levels can modulate Nodal distribution, providing a mechanism for the spatial restriction of Nodal signaling during embryo patterning.

---

## [Decision Letter]

**Decision letter after peer review:**

Thank you for sending your article entitled "Regulation of Nodal signaling propagation by receptor interactions and positive feedback" for peer review at *eLife*. Your article is being evaluated by 3 peer reviewers, and the evaluation is being overseen by a Reviewing Editor and Marianne Bronner as the Senior Editor.

The major concerns of reviewers include:

1. The heavy reliance on morpholinos for loss-of-function studies. The reviewers thought that generation of acvr1b-b mutant and analyses of acvr1b-a/b double mutant would significantly strengthen the conclusion. Likewise, they wondered that while the authors generated or obtained mutant lines for 3 of the 4 type II receptors, redundancy of these genes was not tested by generating double and triple mutants, but by simultaneously knocking down all 4 paralogous genes using MOs. Mutations should be generated for acvr1b-b and acvr2b-b, which should then be tested in combination with other receptor mutations. This is especially important for the type II receptors, because the severe but non-specific phenotypes of quadruple morphant embryos are so difficult to interpret (Figure 3).

2. Overexpression of fluorescently tagged Nodal ligand is not a reliable method for determining signaling range. To make any interpretation of these data, the authors would need to measure how the levels of the injected Nodal compare with endogenous levels and to see how the results vary when varying doses of Nodal are injected,

3. Referring to Figure 6B – the Reviewers wondered how do the timescales for binding and unbinding to receptors compare to that for diffusion? The FRAP dataset requires a clear model for its interpretation.

4. The Reviewers questioned why the quantification of nodal signaling in acvr1b-a and acvr1b-b loss of function and rescue experiments is measured as the number of tiers of cells (distance from ligand source) of pSmad2/3? Instead, for a more accurate measure of signaling level, the amount of pSmad2/3 (fluorescence intensity) should be measured in discreet regions of the embryo.

*Reviewer #1 (Recommendations for the authors):*

While the essential roles of Nodal signaling during early development are well described, the precise function of different zebrafish Nodal receptor paralogs had not been determined. Here, the authors identify 7 genes encoding type I and type II Nodal receptors in zebrafish, 6 of which are maternally deposited and expressed during germ layer specification. Loss-of-function analysis reveals that none of these Nodal receptors is strictly necessary for development, and that loss of both Acvr1b type I receptors is required to yield a Nodal-deficient phenotype. This double loss-of-function was achieved by injecting acvr1b-b morpholino oligonucleotide (MOs) into acvr1b-a mutant embryos, raising the question, why didn't the authors generate mutations in both genes to examine double mutant phenotypes? And while the authors generated or obtained mutant lines for 3 of the 4 type II receptors, they tested for redundancy of these genes not by generating double and triple mutants, but by simultaneously knocking down all 4 paralogous genes using MOs. Because such high doses of MO cause pleiotropic and non-specific defects, the phenotypes of these embryos are difficult to interpret.

Nodal signaling is largely restricted to the margin of pre-gastrulation zebrafish embryos, a gradient that is thought to form through interactions of highly diffusible extracellular Lefty antagonists and less diffusible Nodal ligands. It was recently shown (Lord et al., 2019) that the Oep/Tdgf1 co-receptor restricts Nodal signaling range, and here the authors hypothesized that Nodal type I and II receptors slow the diffusion of Nodal ligands to limit their signaling range. They directly demonstrate that fluorescently tagged Nodal ligands diffuse farther in the absence of both Acvr1b type I receptors and in oep-/- mutants. Using Fluorescence Decay After Photoconversion (FDAP) and Fluorescent Recovery After Photobleaching (FRAP), they determine that Nodal receptors limit ligand range by slowing their diffusion and not through ligand clearance or relay signaling. Namely, FRAP-detected ligand diffusion was found to increase upon loss of type I receptors but remained unchanged upon overexpression of type II receptors, suggesting that type I receptors may play a larger role in ligand distribution (despite the fact that type II receptors are thought to directly and strongly bind Nodal ligands). FDAP experiments revealed no effect on ligand clearance type II receptor overexpression, but it was not examined whether loss of type I receptors alters ligand clearance as it alters ligand diffusion.

Together, these results identify redundancy in Nodal receptor function during early zebrafish development and propose a creative, seemingly paradoxical model in which feed-forward Nodal signaling actually decreases signaling range by boosting expression of receptors that limit ligand diffusion.

1. The authors conclude that, because neither genetic mutations nor translation blocking MOs targeting Nodal receptors produce phenotypes, no genetic compensation is occurring. It would be more convincing to show by qRT-PCR that expression levels of paralogous genes are not altered upon mutation of a receptor gene.

2. Because homozygous loss of any Nodal receptor gene is not lethal, it should be straightforward to generate double mutant embryos and bypass the use of MOs to determine functional redundancy. Mutations should be generated for acvr1b-b and acvr2b-b, which should then be tested in combination with other receptor mutations. This is especially important for the type II receptors, because the severe but non-specific phenotypes of quadruple morphant embryos are so difficult to interpret (Figure 3).

3. The authors state that embryos lacking acvr1b-a or -b alone "had a Nodal signaling range very similar to untreated WT embryos" (line 214), but in Figure 4A these conditions look quite different from one another. Particularly, the pSmad2 signal at the (presumed) dorsal midline looks similar to WT, but appears much dimmer at other positions around the margin of acvr1b LOF embryos. Were the measurement in Figure 4B taken from a single, dorsal region of the margin, which could explain the normal number of pSmad2+ nuclei despite the abnormal appearance of these embryos? Or are these images not representative of the embryos quantified?

4. In figure 5B, it would be nice to see images of Nodal ligand diffusion upon loss of Nodal receptors in addition to loss of oep.

5. Given that type II receptors are reported to interact directly and strongly with Nodal ligands, it is surprising that altering levels of type I receptors affected ligand diffusion while type II levels did not. To test this further, I would like to see measurements of ligand clearance / diffusion upon both loss and gain of function of both type I and type II receptors (rather than only loss of type I and only gain of type II). Because acvr1b mutants exhibit increased diffusion (Figure 6B), I would be interested to know if ligand clearance (stability) is also affected in these mutants? This would help determine if ligand clearance does not actually regulate signaling in this system, or if acvr2b overexpression simply doesn't cause phenotypes (Figures 6A-B).

*Reviewer #2 (Recommendations for the authors):*

Preiß et al., investigate how Nodal receptors regulate Nodal signaling. They first discover several possible new Type I and Type II receptors based on homology to mammalian sequences. They show combined loss of two type I receptors results in a phenotype indicative of loss of nodal signaling. They then perform experiments to suggest that the range of Nodal signaling is regulated by the expression of the type I receptor. The discovery of additional type I receptors is certainly useful, and the phenotype of removing two of them is clear, however, several other claims are not justified by the data. I detail my concerns below.

1. The authors state that Figure 1F shows that inhibition of nodal signaling reduces expression of acvr1b-a, however, I do not see this in the figure where there still appears to be signal in the in situ. If anything, the range of signal appears expanded over the animal pole of the embryo.

2. In Figure 3, the authors show that combining MO against acvr1b-a and acvr1b-b produces a "pinhead" phenotype. Can they speculate why this phenotype is not seen in Figure 3C when knockout of acvr1a-a is combined with MO against acvr1a-b?

3. Figure 3 – in the absence of functional data to support the role of the newly identified type II receptors, given that the loss of function yielded no phenotype, the authors should avoid making a strong claim that these receptors play a role in Nodal signaling.

4. In Figure 4a there seems to be a clear difference in signal intensity between the wildtype and acvr1b-a mutant in the uninflected case. In the mutant much of the margin appears to have reduced or no signal for pSmad2. This should be noted.

5. Figure 5 – results based on overexpression of fluorescently tagged Nodal ligand is not a reliable method for determining signaling range. It is clear from the current study as well as several other that the stoichometry of ligand and receptors will influence the result and so no faith can be put in the results of overexpression. To make any interpretation of this data at all, the authors would need to measure how the levels of the injected Nodal compare with endogenous levels and to see how the results vary when varying doses of Nodal are injected, however, even in this case, much caution is required when drawing conclusions about the range of Nodal in general.

6. Figure 5 – the results shown in C,D are not consistent with what would be expected if binding to the receptor restricted the range of Nodal. In that case, when the receptor is downregulated and Nodal can diffuse more freely, one would expect lower levels near the source but a broader distribution. In this case, there is simply more Nodal everywhere which indicates either the receptor is somehow post-translationally regulating the expression of Nodal or else these embryos simply had more Nodal mRNA injected. Also, quantification is shown for receptor loss of function but not images are shown in B, these should be included.

7. Figure 6 A, B – the authors should show the data from the photoconversion and FRAP experiments and the fit to these curves and then the parameters that were extracted from this.

8. Figure 6A – loss of function for the receptor should be performed as well. If most of the Nodal is already bound, overexpressing the receptor would not be expected to have an effect but loss of function might.

9. Figure 6B – I do not understand the model for diffusion the authors are proposing. How do the timescales for binding and unbinding to receptors compare to that for diffusion? If they are slow, then one expects that the receptor binding would show up in an immobile fraction in the experiment rather than in lowering the diffusion constant. Or are the authors suggesting that the receptor bound Nodal diffuses within the plasma membrane, but more slowly, in which case the data would need to be fit to a two component model. In any event, this FRAP data requires a clear model for its interpretation.

10. Figure 6C, D – quantification is essential for these experiments including quantifying the number of donor cells as well as the induced gradient on pSmad2 in the host.

11. Figure 6C, D – in both cases there only appears to be active signaling immediately adjacent to the graft. This is not consistent with the authors investigation of "medium-range" Nodal signaling. This is important because the authors are trying to investigate whether a relay is essential for this medium range signal, however, if there is no signaling at a distance in the wildtype transplant then it is impossible to draw any conclusion from the transplant into the knockout.

*Reviewer #3 (Recommendations for the authors):*

The nodal signaling pathway plays essential roles in germ-layer induction and patterning during development. While the ligands and co-receptor responsible for nodal activation during zebrafish development have been previously identified, it is still unclear which receptors are required for early zebrafish nodal signaling. Based on loss of function analysis, the authors identify two type I receptors that are redundantly required for nodal signaling during early development. The authors also perform experiments to suggest nodal ligand interactions with the type I receptor can affect ligand diffusion and help shape the nodal signaling gradient.

Strengths:

The identification and characterization of all seven receptors predicted to mediate nodal signaling serves as a nice reference for future studies of this pathway. The genetic loss of function of five of these supports a model where each of the five individually are not required for signaling. The combined loss of function of two type I receptors produces a nodal loss of function phenotype, suggesting that these two function redundantly during early zebrafish development.

The authors present a novel finding that the type I receptor normally functions to limit the diffusion of ligand, which in turn shapes the nodal gradient. This builds upon previous work showing that the nodal co-receptor oep and the nodal antagonist lefty influence the nodal gradient.

The authors show that endogenous levels of nodal ligand can directly activate signaling in cells several cell diameters away without a secondary relay system, which is an important aspect of our understanding of how nodal ligands function in a morphogenetic gradient.

Weaknesses:

Much of the combinatorial receptor loss of function data relies on the use of morpholinos. Given the lack of phenotype of single mutants, the absence of phenotypes in combinatorial loss of function using morpholinos may indicate that those particular receptors are dispensable, or rather that the morpholinos are not efficiently knocking down the gene product, and thus prevents definitive conclusions from being drawn regarding those particular experiments. However the distinctive nodal loss of function phenotype observed in the particular loss of function combination of acvr1b-a (genetic loss of function) and acvr1b-b (morpholino loss of function) provides strong support that these two receptors act redundantly during in zebrafish nodal signaling. These two genes are the central focus of the paper.

The manuscript would be strengthened by adding images for embryos where there are nodal loss or gain of function phenotypes. Although example images of phenotypes are provided, it is not clear what the manipulation was that generated them. In particular there should be images of the acvr1b-a and acvr1b-b combined loss of function phenotype.

The conclusion that acvr1b-a and acvr1b-b function maternally during early zebrafish development needs to be further supported by confirming that the acvr1b-b splice blocking morpholino disrupts splicing of the zygotic transcript.

Suggestions for authors:

Although example images were presented for phenotype classification, it is important to show an example image of the nodal loss of function phenotype generated specifically from the acvr1b-a and acvr1b-b dual loss of function embryos (perhaps in a supplemental figure).

The authors conclude that the maternal transcripts of acvr1b-a and acvr1b-b are critical for nodal signaling based on the strong nodal loss of function phenotype using the acvr1b-b translation blocking MO, and lack of phenotype from the acvr1b-b splice blocking MO. In order to make this conclusion though the authors need to show that the splice blocking MO is actually disrupting the acvr1b-b transcript, as opposed to the lack of phenotype being due to the MO not effectively inhibiting splicing (or causing an in-frame cryptic splice site usage).

Regarding the previous point, something that seems to suggest that the splice blocking acvr1b-b MO is not inhibiting acvr1b-b function is that there is no zygotic nodal loss of function phenotype in the splice-blocking acvr1b-b MO injected acvr1b-a MZ mutants (figure 3C).

It is not clear to me why the quantification of nodal signaling in acvr1b-a and acvr1b-b loss of function and rescue experiments is measured as the number of tiers of cells (distance from ligand source) of pSmad2/3? The ligand distribution should either not be changed or perhaps increased based on receptor loss. Instead, for a more accurate measure of signaling level, the amount of pSmad2/3 (fluorescence intensity) should be measured in discreet regions of the embryo. In the images shown, it appears that the acvr1b-a mutant alone has weaker staining compared to wild-type, which may indicate lower signaling activity even in the absence of a morphological phenotype.

In the FRAP experiment in figure 6B, the example image of acvr1b-a and acvr1b-b loss of function seems to have worse fluorescence recovery than the wild-type embryos, which is opposite of the general trend in the graph of all of the data. When I went to look at the source data to clarify, I just see the final diffusion coefficient numbers that were used to generate the plot. It would be helpful to have the raw fluorescence intensity measurements in this data set.

---

## [Author Response]

The major concerns of reviewers include:1. The heavy reliance on morpholinos for loss-of-function studies. The reviewers thought that generation of acvr1b-b mutant and analyses of acvr1b-a/b double mutant would significantly strengthen the conclusion. Likewise, they wondered that while the authors generated or obtained mutant lines for 3 of the 4 type II receptors, redundancy of these genes was not tested by generating double and triple mutants, but by simultaneously knocking down all 4 paralogous genes using MOs. Mutations should be generated for acvr1b-b and acvr2b-b, which should then be tested in combination with other receptor mutations. This is especially important for the type II receptors, because the severe but non-specific phenotypes of quadruple morphant embryos are so difficult to interpret (Figure 3).

We agree that additional mutants would strengthen our study. As proposed in our revision action plan, we have used a CRISPR knockout strategy to generate combinatorial mutations disrupting both *acvr1* homologs and all four *acvr2* genes. This strategy, recently published by Jason Rihel and colleagues in a breakthrough paper in *eLife*, is “[…] capable of converting >90% of injected embryos directly into F0 biallelic knockouts” and “[…] sufficiently robust to knockout multiple genes in the same animal […]” (https://elifesciences.org/articles/59683, Kroll et al., 2021).

For the Type I receptors Acvr1b-a and Acvr1b-b, the new CRISPR knockouts validated the specificity of our previous experiments, showing that the receptors redundantly mediate Nodal signaling (new Figure 3). Using the same approach, we found that Acvr2 receptors function partly redundantly to pattern the early zebrafish embryo, do not fulfill the function of classical Type II Nodal receptors and act at least partially independently of Nodal (new Figure 2, new Figure 2 —figure supplement 2).

In the revised manuscript, we have now used a total of three different methods to generate combinatorial loss-of-function conditions for *acvr1b-a*/*acvr1b-b* and *acvr2a-a*/*acvr2a-b*/*acvr2b-a*/*acvr2b-b*: (1) Morpholino-mediated knockdown, (2) CRISPR knockouts and (3) stable mutants in combination with the former two techniques. Using these different methods we obtained the same results, strengthening our conclusions. For example, the phenotypes observed upon quadruple *acvr2* gene knockout (new Figure 2A,B) were recapitulated by both the quadruple knockdown and the knockout of *acvr2b-b* in triple mutants homozygous for *acvr2a-a*^*-/-*^, *acvr2a-b*^*-/-*^ and *acvr2b-a*^*-/-*^ (new Figure 2 —figure supplement 2A-C).

2. Overexpression of fluorescently tagged Nodal ligand is not a reliable method for determining signaling range. To make any interpretation of these data, the authors would need to measure how the levels of the injected Nodal compare with endogenous levels and to see how the results vary when varying doses of Nodal are injected,

We agree that it is important to ensure that endogenous receptors are not saturated by ectopic fluorescently tagged Nodal in gradient formation assays. Unfortunately, endogenous Nodal levels are unknown and currently impossible to determine precisely due to the lack of appropriate antibodies and transgenic lines. In order to visualize the distribution of Nodal signals in zebrafish embryos, expression of fluorescently tagged proteins by mRNA injection is currently the best approach and well recapitulates the endogenous signaling range and dynamics (e.g. Müller et al., 2012; Soh et al., 2020; Kuhn et al., 2022; also see Stapornwongkul and Briscoe 2022). Fluorescently tagged Nodal ligands expressed from localized ectopic sources can even induce the formation of a full secondary axis in zebrafish embryos, demonstrating that these fusion proteins have full biological activity and relevant signaling ranges to orchestrate complex morphogenetic processes (Soh et al., 2020). In our gradient formation assays (new Figure 5, New Figure 5 —figure supplement 1), we observe clear differences in the distribution of Nodal-GFP in receptor mutants compared to wild-type embryos, indicating that Nodal-GFP expression does not lead to saturation of endogenous Nodal receptors.

To further validate the physiological relevance of the Nodal-GFP levels in our gradient formation assays, we have now performed additional experiments and injected different amounts of mRNA encoding Nodal-GFP into maternal-zygotic *squint*;*cyclops* mutants devoid of any endogenous Nodal source (new Figure 5 —figure supplement 1E). Importantly, the amount of mRNA that we had used in the YSL gradient formation assay rescued the mutant *squint*;*cyclops* phenotype the best, suggesting that the amount of Nodal-GFP in our gradient formation experiments is near physiological levels.

3. Referring to Figure 6B – the Reviewers wondered how do the timescales for binding and unbinding to receptors compare to that for diffusion? The FRAP dataset requires a clear model for its interpretation.

We assume rapid binding and relatively slow unbinding of Nodal ligands and receptors, and in the revised manuscript we now refer to our previous theoretical and experimental studies that provide detailed models (Müller et al., 2012; Müller et al., 2013; Bläßle et al., 2018; Mörsdorf and Müller 2019; Kuhn et al., 2022). Specifically, in our recent single-molecule tracking experiments (Kuhn et al., 2022), we have measured binding times of 16 s for fluorescently tagged Cyclops and 11 s for fluorescently tagged Squint in zebrafish embryos on a nanometer-to-micrometer scale. These experimental measurements are in remarkable agreement with our previous FRAP-based tissue-scale binding time predictions of 18 s and 4 s for Cyclops-GFP and Squint-GFP, respectively (Müller et al., 2012). Importantly, these binding times can directly explain how the effective diffusion timescale of Nodal in FRAP experiments arises from local interactions with immobile receptors (Müller et al., 2012; Müller et al., 2013; Bläßle et al., 2018; Mörsdorf and Müller 2019; Kuhn et al., 2022).

4. The Reviewers questioned why the quantification of nodal signaling in acvr1b-a and acvr1b-b loss of function and rescue experiments is measured as the number of tiers of cells (distance from ligand source) of pSmad2/3? Instead, for a more accurate measure of signaling level, the amount of pSmad2/3 (fluorescence intensity) should be measured in discreet regions of the embryo.

Thank you for the suggestion. We originally performed the quantification of pSmad2/3 signal as a function of cell tiers from the margin since the number of pSmad2/3-positive nuclei can be easily manually counted and the approach is well accepted in the field (e.g. van Boxtel et al., 2015). To corroborate our findings, we have now complemented our analyses with the suggested measurements of pSmad2/3 fluorescence intensities in discrete regions of the embryo. Specifically, we performed line-average measurements of pSmad2/3 distributions in lateral and dorsal domains, and the resulting quantifications confirmed our previous conclusions (new Figure 4 —figure supplement 1, Figure 4).

Reviewer #1 (Recommendations for the authors):While the essential roles of Nodal signaling during early development are well described, the precise function of different zebrafish Nodal receptor paralogs had not been determined. Here, the authors identify 7 genes encoding type I and type II Nodal receptors in zebrafish, 6 of which are maternally deposited and expressed during germ layer specification. Loss-of-function analysis reveals that none of these Nodal receptors is strictly necessary for development, and that loss of both Acvr1b type I receptors is required to yield a Nodal-deficient phenotype. This double loss-of-function was achieved by injecting acvr1b-b morpholino oligonucleotide (MOs) into acvr1b-a mutant embryos, raising the question, why didn't the authors generate mutations in both genes to examine double mutant phenotypes? And while the authors generated or obtained mutant lines for 3 of the 4 type II receptors, they tested for redundancy of these genes not by generating double and triple mutants, but by simultaneously knocking down all 4 paralogous genes using MOs. Because such high doses of MO cause pleiotropic and non-specific defects, the phenotypes of these embryos are difficult to interpret.

We agree that additional mutants would strengthen our study. As proposed in our revision action plan, we have used a CRISPR knockout strategy to generate combinatorial mutations disrupting both *acvr1* homologs and all four *acvr2* genes. This strategy, recently published by Jason Rihel and colleagues in a breakthrough paper in *eLife*, is “[…] capable of converting >90% of injected embryos directly into F0 biallelic knockouts” and “[…] sufficiently robust to knockout multiple genes in the same animal […]” (https://elifesciences.org/articles/59683, Kroll et al., 2021).

For the Type I receptors Acvr1b-a and Acvr1b-b, the new CRISPR knockouts validated the specificity of our previous experiments, showing that the receptors redundantly mediate Nodal signaling (new Figure 3). Using the same approach, we found that Acvr2 receptors function partly redundantly to pattern the early zebrafish embryo, do not fulfill the function of classical Type II Nodal receptors and act at least partially independently of Nodal (new Figure 2, new Figure 2 —figure supplement 2).

In the revised manuscript, we have now used a total of three different methods to generate combinatorial loss-of-function conditions for *acvr1b-a*/*acvr1b-b* and *acvr2a-a*/*acvr2a-b*/*acvr2b-a*/*acvr2b-b*: (1) Morpholino-mediated knockdown, (2) CRISPR knockouts and (3) stable mutants in combination with the former two techniques. Using these different methods we obtained the same results, strengthening our conclusions. For example, the phenotypes observed upon quadruple *acvr2* gene knockout (new Figure 2A,B) were recapitulated by both the quadruple knockdown and the knockout of *acvr2b-b* in triple mutants homozygous for *acvr2a-a*^*-/-*^, *acvr2a-b*^*-/-*^ and *acvr2b-a*^*-/-*^ (new Figure 2 —figure supplement 2A-C).

Nodal signaling is largely restricted to the margin of pre-gastrulation zebrafish embryos, a gradient that is thought to form through interactions of highly diffusible extracellular Lefty antagonists and less diffusible Nodal ligands. It was recently shown (Lord et al., 2019) that the Oep/Tdgf1 co-receptor restricts Nodal signaling range, and here the authors hypothesized that Nodal type I and II receptors slow the diffusion of Nodal ligands to limit their signaling range. They directly demonstrate that fluorescently tagged Nodal ligands diffuse farther in the absence of both Acvr1b type I receptors and in oep-/- mutants. Using Fluorescence Decay After Photoconversion (FDAP) and Fluorescent Recovery After Photobleaching (FRAP), they determine that Nodal receptors limit ligand range by slowing their diffusion and not through ligand clearance or relay signaling. Namely, FRAP-detected ligand diffusion was found to increase upon loss of type I receptors but remained unchanged upon overexpression of type II receptors, suggesting that type I receptors may play a larger role in ligand distribution (despite the fact that type II receptors are thought to directly and strongly bind Nodal ligands). FDAP experiments revealed no effect on ligand clearance type II receptor overexpression, but it was not examined whether loss of type I receptors alters ligand clearance as it alters ligand diffusion.

In the revised manuscript, we now include FDAP measurements in *acvr1b-a;acvr1b-b* loss-of-function conditions (Figure 6A). We found that the absence of the Type I receptors does not cause a decrease in the clearance rate of Squint-GFP or Cyclops-GFP, showing that the broadening of the gradients (new Figure 5, new Figure 5 —figure supplement 1) cannot be explained through increased protein stability.

Together, these results identify redundancy in Nodal receptor function during early zebrafish development and propose a creative, seemingly paradoxical model in which feed-forward Nodal signaling actually decreases signaling range by boosting expression of receptors that limit ligand diffusion.1. The authors conclude that, because neither genetic mutations nor translation blocking MOs targeting Nodal receptors produce phenotypes, no genetic compensation is occurring. It would be more convincing to show by qRT-PCR that expression levels of paralogous genes are not altered upon mutation of a receptor gene.

We apologize for the confusion. We did not want to claim that no genetic compensation is occurring, but argue that morpholino-mediated knockdown on the mRNA level could help to reveal effects masked by genetic compensation in the mutants. As suggested, we have now included qRT-PCR experiments in the revised manuscript (new Figure 2 —figure supplement 1E,F and new Figure 3 —figure supplement 1D). The results suggest that in some instances genetic compensation might indeed occur. We now refer to these new results in the main text to more clearly motivate the use of morpholinos as tools that may reveal phenotypes that could be masked by genetic compensation.

2. Because homozygous loss of any Nodal receptor gene is not lethal, it should be straightforward to generate double mutant embryos and bypass the use of MOs to determine functional redundancy. Mutations should be generated for acvr1b-b and acvr2b-b, which should then be tested in combination with other receptor mutations. This is especially important for the type II receptors, because the severe but non-specific phenotypes of quadruple morphant embryos are so difficult to interpret (Figure 3).

As detailed above, we have executed the suggested experiments to corroborate our findings.

3. The authors state that embryos lacking acvr1b-a or -b alone "had a Nodal signaling range very similar to untreated WT embryos" (line 214), but in Figure 4A these conditions look quite different from one another. Particularly, the pSmad2 signal at the (presumed) dorsal midline looks similar to WT, but appears much dimmer at other positions around the margin of acvr1b LOF embryos. Were the measurement in Figure 4B taken from a single, dorsal region of the margin, which could explain the normal number of pSmad2+ nuclei despite the abnormal appearance of these embryos? Or are these images not representative of the embryos quantified?

In Figure 4B, we indeed measured Nodal signaling range on the dorsal side. We have now stated this more clearly in the figure legend and in the main text of the revised manuscript. Moreover, we have complemented our analyses with pSmad2/3 intensity quantifications in dorsal and lateral regions (new Figure 4 —figure supplement 1). These new analyses confirmed that embryos lacking Acvr1b-a or Acvr1b-b have a similar Nodal signaling range as wild-type embryos in dorsal regions. In lateral domains, at a distance of more than 30 µm from the margin, the pSmad2/3 intensities are slightly lower in *acvr1b-a^-/-^* compared to wild-type embryos (new Figure 4 —figure supplement 1). We have noted this in the text and selected example images better representing these results.

4. In figure 5B, it would be nice to see images of Nodal ligand diffusion upon loss of Nodal receptors in addition to loss of oep.

Thank you, we have included these images in the revised manuscript (new Figure 5 —figure supplement 1D).

5. Given that type II receptors are reported to interact directly and strongly with Nodal ligands, it is surprising that altering levels of type I receptors affected ligand diffusion while type II levels did not. To test this further, I would like to see measurements of ligand clearance / diffusion upon both loss and gain of function of both type I and type II receptors (rather than only loss of type I and only gain of type II). Because acvr1b mutants exhibit increased diffusion (Figure 6B), I would be interested to know if ligand clearance (stability) is also affected in these mutants? This would help determine if ligand clearance does not actually regulate signaling in this system, or if acvr2b overexpression simply doesn't cause phenotypes (Figures 6A-B).

We have included FDAP measurements for Squint-Dendra2 and Cyclops-Dendra2 in Acvr1 loss-of-function embryos in our revised manuscript (new Figure 6A). Our results show that the broadening of the Nodal gradient observed upon loss of the Type I receptors (Figure 5) cannot be explained by increased stability of the Nodal proteins (new Figure 6A).

For Acvr2, the new combinatorial knockout experiments suggest that the receptors function partly redundantly and at least partially independently of Nodal (new Figure 2, new Figure 2 —figure supplement 2), requiring a set of additional experiments beyond the scope of the present work. We will therefore elucidate the mechanism-of-action and precise function of Acvr2 receptors in a follow-up study.

Reviewer #2 (Recommendations for the authors):Preiß et al., investigate how Nodal receptors regulate Nodal signaling. They first discover several possible new Type I and Type II receptors based on homology to mammalian sequences. They show combined loss of two type I receptors results in a phenotype indicative of loss of nodal signaling. They then perform experiments to suggest that the range of Nodal signaling is regulated by the expression of the type I receptor. The discovery of additional type I receptors is certainly useful, and the phenotype of removing two of them is clear, however, several other claims are not justified by the data. I detail my concerns below.1. The authors state that Figure 1F shows that inhibition of nodal signaling reduces expression of acvr1b-a, however, I do not see this in the figure where there still appears to be signal in the in situ. If anything, the range of signal appears expanded over the animal pole of the embryo.

We apologize for the poor quality of the staining in the previous picture, where the apparent background signal likely resulted from overstaining. To corroborate our findings, we have now carefully repeated the in situ staining and replaced the respective image in Figure 1F. The new image demonstrates the lack of signal more clearly, recapitulating the strong downregulation of *acvr1b-a* upon Nodal signaling inhibition determined by qRT-PCR (Figure 1E).

2. In Figure 3, the authors show that combining MO against acvr1b-a and acvr1b-b produces a "pinhead" phenotype. Can they speculate why this phenotype is not seen in Figure 3C when knockout of acvr1a-a is combined with MO against acvr1a-b?

The pinhead phenotype in *acvr1b-a*;*acvr1b-b* morphants is reminiscent of the phenotype in zygotic *oep* mutants, whereas the full Nodal loss-of-function phenotype in *acvr1b-a* mutants injected with *acvr1b-b* morpholino is similar to the phenotype in maternal-zygotic *oep* mutants (Gritsman et al., 1999). This suggests that the *acvr1b-a* morpholino does not fully abolish Acvr1b-a activity. We have included this speculation in the revised manuscript as follows:

“Morpholino mediated double knockdown of acvr1b-a and acvr1b-b resulted in a clear loss of head mesoderm at 1 dpf, leading to the distinctive fused-eye phenotype and a curved body axis associated with loss of Nodal signaling (Figure 3 —figure supplement 2A,B). However, somites still formed in the trunk region, suggesting an incomplete loss of Nodal signaling possibly due to maternal deposition of receptor proteins or incomplete mRNA knockdown”.

3. Figure 3 – in the absence of functional data to support the role of the newly identified type II receptors, given that the loss of function yielded no phenotype, the authors should avoid making a strong claim that these receptors play a role in Nodal signaling.

We fully agree. We have now extended our analysis of the putative Type II receptors *acvr2a-a*, *acvr2a-b*, *acvr2b-a* and *acvr2b-b* by generating quadruple knockout embryos (new Figure 2A,B) and by combining the knockout of *acvr2b-b* with triple homozygous *acvr2a-a^-/-^, acvr2a-b^-/-^* and *acvr2b-a^-/-^* mutants (new Figure 2 —figure supplement 2A-C). The morphological and molecular phenotypes that we observe in these embryos show that receptors function partly redundantly to pattern the early zebrafish embryo and that they do so at least partially independently of Nodal. In the revised manuscript we now emphasize that the receptors do not fulfill the function of classical Type II Nodal receptors.

4. In Figure 4a there seems to be a clear difference in signal intensity between the wildtype and acvr1b-a mutant in the uninflected case. In the mutant much of the margin appears to have reduced or no signal for pSmad2. This should be noted.

This was also noticed by Reviewer #1 in specific point 3. As suggested, we now point out more clearly that we measured the number of pSmad2/3-positive cell tiers only on the dorsal side.

Furthermore, we have included measurements in lateral regions in addition to the dorsal domain (new Figure 4 —figure supplement 1). In lateral regions, *acvr1b‑a^-^*^/*-*^ mutants showed a slight reduction in pSmad2/3 signal intensity at distances above 30 µm from the margin compared to wild-type embryos. We have noted this in the revised manuscript as suggested.

5. Figure 5 – results based on overexpression of fluorescently tagged Nodal ligand is not a reliable method for determining signaling range. It is clear from the current study as well as several other that the stoichometry of ligand and receptors will influence the result and so no faith can be put in the results of overexpression. To make any interpretation of this data at all, the authors would need to measure how the levels of the injected Nodal compare with endogenous levels and to see how the results vary when varying doses of Nodal are injected, however, even in this case, much caution is required when drawing conclusions about the range of Nodal in general.

We agree that it is important to ensure that endogenous receptors are not saturated by ectopic fluorescently tagged Nodal in gradient formation assays. Unfortunately, endogenous Nodal levels are unknown and currently impossible to determine precisely due to the lack of appropriate antibodies and transgenic lines. In order to visualize the distribution of Nodal signals in zebrafish embryos, expression of fluorescently tagged proteins by mRNA injection is currently the best approach and well recapitulates the endogenous signaling range and dynamics (e.g. Müller et al., 2012; Soh et al., 2020; Kuhn et al., 2022; also see Stapornwongkul and Briscoe 2022). Fluorescently tagged Nodal ligands expressed from localized ectopic sources can even induce the formation of a full secondary axis in zebrafish embryos, demonstrating that these fusion proteins have full biological activity and relevant signaling ranges to orchestrate complex morphogenetic processes (Soh et al., 2020). In our gradient formation assays (new Figure 5, New Figure 5 —figure supplement 1), we observe clear differences in the distribution of Nodal-GFP in receptor mutants compared to wild-type embryos (Figure 5, Figure 5 —figure supplement 1), indicating that Nodal-GFP expression does not lead to saturation of endogenous Nodal receptors.

To further validate the physiological relevance of the Nodal-GFP levels in our gradient formation assays, we have now performed additional experiments and injected different amounts of mRNA encoding Nodal-GFP into maternal-zygotic *squint*;*cyclops* mutants devoid of any endogenous Nodal source (new Figure 5 —figure supplement 1E). Importantly, the amount of mRNA that we had used in the YSL gradient formation assay rescued the mutant *squint*;*cyclops* phenotype the best, suggesting that the amount of Nodal-GFP in our gradient formation experiments is near physiological levels.

6. Figure 5 – the results shown in C,D are not consistent with what would be expected if binding to the receptor restricted the range of Nodal. In that case, when the receptor is downregulated and Nodal can diffuse more freely, one would expect lower levels near the source but a broader distribution. In this case, there is simply more Nodal everywhere which indicates either the receptor is somehow post-translationally regulating the expression of Nodal or else these embryos simply had more Nodal mRNA injected. Also, quantification is shown for receptor loss of function but not images are shown in B, these should be included.

To account for differences in fluorescence intensities in the YSL gradient formation assay possibly caused by differences in microscopy settings, laser power or injected Nodal mRNA amounts, we have now calculated normalized signal distributions (new Figure 5 —figure supplement 1C). As also suggested by Reviewer #1, we have additionally included images of Squint-GFP and Cyclops-GFP in receptor loss-of-function conditions in the revised manuscript (new Figure 5 —figure supplement 1B).

To corroborate our findings of broadened gradients when the receptor is downregulated, we have performed additional experiments and measured the distributions of Squint-GFP and Cyclops-GFP expressed from localized cell clones as an alternative to the YSL gradient formation assay (new Figure 5). In agreement with our previous results, the distributions of Squint-GFP and Cyclops-GFP were broadened in the absence of the Nodal Type I receptors and co-receptors, consistent with the increased effective diffusivity measured by FRAP experiments (Figure 6B).

7. Figure 6 A, B – the authors should show the data from the photoconversion and FRAP experiments and the fit to these curves and then the parameters that were extracted from this.

Thank you for this suggestion. We have included representative fits and raw data for the FDAP and FRAP experiments in Figure 6 —figure supplement 1.

8. Figure 6A – loss of function for the receptor should be performed as well. If most of the Nodal is already bound, overexpressing the receptor would not be expected to have an effect but loss of function might.

We agree with the reviewer and have included these additional FDAP experiments in the revised manuscript (new Figure 6A). The results demonstrate that the absence of the Type I receptors does not cause a decrease in the clearance rate of Squint-GFP or Cyclops-GFP, showing that the broadening of the gradients (Figure 5) cannot be explained through increased protein stability.

9. Figure 6B – I do not understand the model for diffusion the authors are proposing. How do the timescales for binding and unbinding to receptors compare to that for diffusion? If they are slow, then one expects that the receptor binding would show up in an immobile fraction in the experiment rather than in lowering the diffusion constant. Or are the authors suggesting that the receptor bound Nodal diffuses within the plasma membrane, but more slowly, in which case the data would need to be fit to a two component model. In any event, this FRAP data requires a clear model for its interpretation.

We assume rapid binding and relatively slow unbinding of Nodal ligands and receptors, and in the revised manuscript we now refer to our previous theoretical and experimental studies that provide detailed models (Müller et al., 2012; Müller et al., 2013; Bläßle et al., 2018; Mörsdorf and Müller 2019; Kuhn et al., 2022).

Specifically, we used the simple synthesis-diffusion-clearance model

∂c∂t=D∇2−k1c+k2 with our previously measured clearance rate constants *k_1_* for Squint and Cyclops (Müller et al., 2012), uniform production *k_2_* and the effective diffusion coefficient *D* as the fitting parameter for the time-dependent recovery of the fluorescence intensity *c* (Müller et al., 2012; Pomreinke et al., 2017; Bläßle et al., 2018; Mörsdorf and Müller 2019). It can be shown that the effective diffusion coefficient *D* of a molecule is equivalent to D=Dfree[Receptor]koffkon+1 when binding of a ligand with the local free diffusion coefficient *D*_free_ to an immobile receptor with the on-rate *k*_on_ and the off-rate *k*_off_ is fast and reversible (Crank 1979; Sprague et al., 2004; Thorne et al., 2008; Miura et al., 2009; Müller et al., 2012; Müller et al., 2013; Bläßle et al., 2018; Mörsdorf and Müller 2019).

In our recent single-molecule tracking experiments (Kuhn et al., 2022), we have measured binding times of 16 s for fluorescently tagged Cyclops and 11 s for fluorescently tagged Squint in zebrafish embryos on a local nanometer-to-micrometer scale. These experimental measurements are in remarkable agreement with our previous FRAP-based tissue-scale binding time predictions of 18 s and 4 s for Cyclops-GFP and Squint-GFP, respectively (Müller et al., 2012). Importantly, these binding times can directly explain how the effective diffusion timescale of Nodal in FRAP experiments arises from local interactions with immobile receptors (Müller et al., 2012; Müller et al., 2013; Bläßle et al., 2018; Mörsdorf and Müller 2019; Kuhn et al., 2022).

10. Figure 6C, D – quantification is essential for these experiments including quantifying the number of donor cells as well as the induced gradient on pSmad2 in the host.

We have included these suggested quantifications in the revised paper (new Figure 6 —figure supplement 2A-C). We found that the number of transplanted donor cells was not significantly different between wild-type and MZ*sqt*^*-/-*^;*cyc*^*-/-*^ hosts (see Materials and methods – Statistical analysis in the revised manuscript). In addition, the pSmad2/3 gradient normalized to the number of transplanted cells was broader and of higher intensity in MZ*sqt*^*-/-*^;*cyc*^*-/-*^ compared to wild-type host embryos, confirming our previous conclusions.

11. Figure 6C, D – in both cases there only appears to be active signaling immediately adjacent to the graft. This is not consistent with the authors investigation of "medium-range" Nodal signaling. This is important because the authors are trying to investigate whether a relay is essential for this medium range signal, however, if there is no signaling at a distance in the wildtype transplant then it is impossible to draw any conclusion from the transplant into the knockout.

The idea that the spreading of endogenous Nodal signaling requires relay is widely discussed in the current literature (e.g. van Boxtel et al., 2015; Rogers and Müller 2019; Liu et al., 2022). In these relay models, Nodal has at best a short diffusion range and requires a chain of signaling between adjacent cells to act over longer distances. For example, the Warmflash lab has recently stated that in an open micropatterned hESC-based 2D gastruloid system “Nodal does not travel more than one cell diameter in any context we examined” (Liu et al., 2022). This contrasts with the considerable number of pSmad2/3 positive nuclei (on average 4 and up to 15) that we found outside of margin-derived transplants in wild-type embryos and the even higher number (on average 8 and up to 31) in *MZsqt^-/-^;cyc^-/-^* hosts (Figure 6 —figure supplement 2E). Together with the longer Nodal signaling range in *MZsqt^-/-^;cyc^-/-^* hosts (Figure 6 —figure supplement 2B) this demonstrates that in an embryonic context Nodal can indeed act directly over a distance.

Reviewer #3 (Recommendations for the authors):The nodal signaling pathway plays essential roles in germ-layer induction and patterning during development. While the ligands and co-receptor responsible for nodal activation during zebrafish development have been previously identified, it is still unclear which receptors are required for early zebrafish nodal signaling. Based on loss of function analysis, the authors identify two type I receptors that are redundantly required for nodal signaling during early development. The authors also perform experiments to suggest nodal ligand interactions with the type I receptor can affect ligand diffusion and help shape the nodal signaling gradient.Strengths:The identification and characterization of all seven receptors predicted to mediate nodal signaling serves as a nice reference for future studies of this pathway. The genetic loss of function of five of these supports a model where each of the five individually are not required for signaling. The combined loss of function of two type I receptors produces a nodal loss of function phenotype, suggesting that these two function redundantly during early zebrafish development.The authors present a novel finding that the type I receptor normally functions to limit the diffusion of ligand, which in turn shapes the nodal gradient. This builds upon previous work showing that the nodal co-receptor oep and the nodal antagonist lefty influence the nodal gradient.The authors show that endogenous levels of nodal ligand can directly activate signaling in cells several cell diameters away without a secondary relay system, which is an important aspect of our understanding of how nodal ligands function in a morphogenetic gradient.

We thank the reviewer for appreciating the strengths of our study.

Weaknesses:Much of the combinatorial receptor loss of function data relies on the use of morpholinos. Given the lack of phenotype of single mutants, the absence of phenotypes in combinatorial loss of function using morpholinos may indicate that those particular receptors are dispensable, or rather that the morpholinos are not efficiently knocking down the gene product, and thus prevents definitive conclusions from being drawn regarding those particular experiments. However the distinctive nodal loss of function phenotype observed in the particular loss of function combination of acvr1b-a (genetic loss of function) and acvr1b-b (morpholino loss of function) provides strong support that these two receptors act redundantly during in zebrafish nodal signaling. These two genes are the central focus of the paper.The manuscript would be strengthened by adding images for embryos where there are nodal loss or gain of function phenotypes.

As suggested, we have included these images in our revised manuscript (new Figure 2 —figure supplement 2, new Figure 3A, new Figure 5 —figure supplement 1E).

Although example images of phenotypes are provided, it is not clear what the manipulation was that generated them. In particular there should be images of the acvr1b-a and acvr1b-b combined loss of function phenotype.

We have included additional more detailed example images for the individual conditions in our revised manuscript and more clearly pointed out what the manipulation was that generated them (new Figure 2A and new Figure 3A).

The conclusion that acvr1b-a and acvr1b-b function maternally during early zebrafish development needs to be further supported by confirming that the acvr1b-b splice blocking morpholino disrupts splicing of the zygotic transcript.

We have performed the suggested analysis and included the results in our revised manuscript (new Figure 3 —figure supplement 2D). We found that the *acvr1b-b* splice-blocking morpholino indeed disrupts the splicing of the zygotic transcript but does not affect the already spliced maternal transcript. Additionally, we show that the phenotype observed upon combinatorial zygotic knockout of *acvr1b-a* and *acvr1b-b* recapitulates the zygotic knockout of *oep* (new Figure 3A,C), further supporting the idea that maternally deposited receptor mRNAs contribute to Nodal signaling.

Suggestions for authors:Although example images were presented for phenotype classification, it is important to show an example image of the nodal loss of function phenotype generated specifically from the acvr1b-a and acvr1b-b dual loss of function embryos (perhaps in a supplemental figure).

We have included additional more detailed example images for the individual conditions in our revised manuscript for clarification (new Figure 3A).

The authors conclude that the maternal transcripts of acvr1b-a and acvr1b-b are critical for nodal signaling based on the strong nodal loss of function phenotype using the acvr1b-b translation blocking MO, and lack of phenotype from the acvr1b-b splice blocking MO. In order to make this conclusion though the authors need to show that the splice blocking MO is actually disrupting the acvr1b-b transcript, as opposed to the lack of phenotype being due to the MO not effectively inhibiting splicing (or causing an in-frame cryptic splice site usage).

Thank you for the suggestion. We have performed the suggested analysis and included the results in our revised manuscript (new Figure 3 —figure supplement 2D). We found that the *acvr1b-b* splice-blocking morpholino indeed disrupts the splicing of the zygotic transcript but does not affect the already spliced maternal transcript. However, even with 2 ng of injected morpholino, spliced transcripts are detected at shield stage (new Figure 3 —figure supplement 2D). This suggests either a long persistence of the maternal transcripts or incomplete knockdown.

To further test whether maternally deposited receptor mRNA contributes to Nodal signaling, we have complemented our analysis with zygotic combinatorial receptor mutants (new Figure 3A,C). We show that the phenotypes observed upon zygotic knockout of both *acvr1b-a* and *acvr1b-b* resemble the zygotic *oep* knockout phenotype*,* indicating a partial rescue of the Nodal loss-of-function phenotype by maternal transcripts.

Regarding the previous point, something that seems to suggest that the splice blocking acvr1b-b MO is not inhibiting acvr1b-b function is that there is no zygotic nodal loss of function phenotype in the splice-blocking acvr1b-b MO injected acvr1b-a MZ mutants (figure 3C).

We fully agree with this interpretation. As outlined above, we did observe a disruption of transcript splicing upon injection of the morpholino, but we also observed remaining spliced transcript at shield stage indicative of persistent maternal transcripts or incomplete knockdown. This might explain the absence of a zygotic loss-of-function phenotype. However, together with the results obtained from the combinatorial zygotic receptor mutants (see above), this suggests a contribution of maternal receptor mRNA. We have adjusted our interpretations in the text accordingly.

It is not clear to me why the quantification of nodal signaling in acvr1b-a and acvr1b-b loss of function and rescue experiments is measured as the number of tiers of cells (distance from ligand source) of pSmad2/3? The ligand distribution should either not be changed or perhaps increased based on receptor loss. Instead, for a more accurate measure of signaling level, the amount of pSmad2/3 (fluorescence intensity) should be measured in discreet regions of the embryo. In the images shown, it appears that the acvr1b-a mutant alone has weaker staining compared to wild-type, which may indicate lower signaling activity even in the absence of a morphological phenotype.

This was also suggested by Reviewer #1 and Reviewer #2 (see above), and we have included additional quantifications in dorsal and lateral domains in the revised manuscript (new Figure 4 —figure supplement 1).

In the FRAP experiment in figure 6B, the example image of acvr1b-a and acvr1b-b loss of function seems to have worse fluorescence recovery than the wild-type embryos, which is opposite of the general trend in the graph of all of the data. When I went to look at the source data to clarify, I just see the final diffusion coefficient numbers that were used to generate the plot. It would be helpful to have the raw fluorescence intensity measurements in this data set.

As suggested, we have included representative fits and raw fluorescence intensity measurements in the revised manuscript (Figure 6 —figure supplement 1A, Figure 6 —figure supplement 1 – source data).